# Scaling Large Vision-Language Model RL Training via Efficient Load Balancing

**Zerui Wang**[1,2]**, Qinghao Hu**[3]**, Chang Chen**[5]**, Jiecheng Zhou**[4,2]
**Haojie Duanmu**[1]**, Xingcheng Zhang**[2]**, Peng Sun**[7]**, Dahua Lin**[6,2]
[1]Shanghai Jiao Tong University    [2]Shanghai Artificial Intelligence Laboratory
[3]Massachusetts Institute of Technology    [4]University of Science and Technology of China
[5]Peking University    [6]The Chinese University of Hong Kong    [7]Unaffiliated

## Abstract

Reinforcement learning (RL) is increasingly used to align vision–language models (VLMs), yet scaling RL for VLMs is bottlenecked by multimodal data handling and extreme workload skew. In typical RL pipelines, visual data loading and preprocessing are centralized, creating severe I/O and CPU/memory stragglers, while batches that mix short image-text prompts with long video contexts lead to large cross-GPU imbalance during rollouts, inference, and training. We present FlexRL, an end-to-end system that removes these bottlenecks. FlexRL introduces: (1) **ShadowLoader**, a distributed, metadata-driven pipeline that keeps only lightweight visual metadata on the controller, pushes decoding and preprocessing to worker-side preprocessors, and asynchronously materializes tensors to overlap I/O with GPU computation; (2) **FlexUlysses**, a cost-aware sub-sequence sharding and execution engine that adaptively splits sequences to balance compute and memory. Our evaluation shows that across multiple VLM scales and multimodal datasets on 128-GPU clusters, FlexRL improves end-to-end throughput by up to $8.47\times$ over state-of-the-art RL systems.

## 1    Introduction

Reinforcement learning (RL) has proven to be a powerful paradigm for aligning vision–language models (VLMs) with human preferences and enhancing their instruction-following capabilities (Team et al., 2025a;b;d; Wang et al., 2025b; Cha, 2023; Hu et al., 2024b; Zhou et al., 2025). A typical RL workflow for VLMs involves several distinct stages: (1) loading diverse multimodal data and prompts, (2) generating responses via policy model inference (i.e., rollouts), and (3) updating the model parameters, etc. While conceptually straightforward, scaling this process for VLMs exposes severe system-level bottlenecks across the entire pipeline, hindering efficiency and scalability.

RL training of VLMs suffers from two types of load imbalance: **(1) Vision data imbalance.** Firstly, the data loading phase becomes a significant bottleneck. VLMs are trained on heterogeneous datasets containing text, high-resolution images, and video clips. In many RL frameworks (e.g., VeRL (Sheng et al., 2025)), data loading and preprocessing are centralized, causing the master node to become a straggler, limited by its memory and compute capacity, especially when handling massive vision data. **(2) Model execution imbalance.** Secondly, the model execution phase suffers from a critical load imbalance problem. A single batch of sequences can contain a mix of short image-text queries, long text-only reasoning tasks, and video inputs with tens of thousands of tokens. This extreme variation in sequence length and modality leads to a highly skewed distribution of computational and memory loads across GPUs. Consequently, some devices are overwhelmed while others remain underutilized, creating a bottleneck that stalls the entire distributed system.

Existing systems fail to provide a holistic solution for the VLM RL pipeline. For the vision data imbalance, most RL frameworks (Sheng et al., 2025; Hu et al., 2024a) are optimized for text-only models and lack sophisticated mechanisms to handle the load imbalance inherent in multimodal data. These frameworks typically employ naive sequence bucketing and packing strategies, which fail to tackle highly imbalanced workloads. On the other hand, general-purpose large model training systems (Wang et al., 2025c;d; Li et al., 2024; Ge et al., 2024; 2025) propose heterogeneous

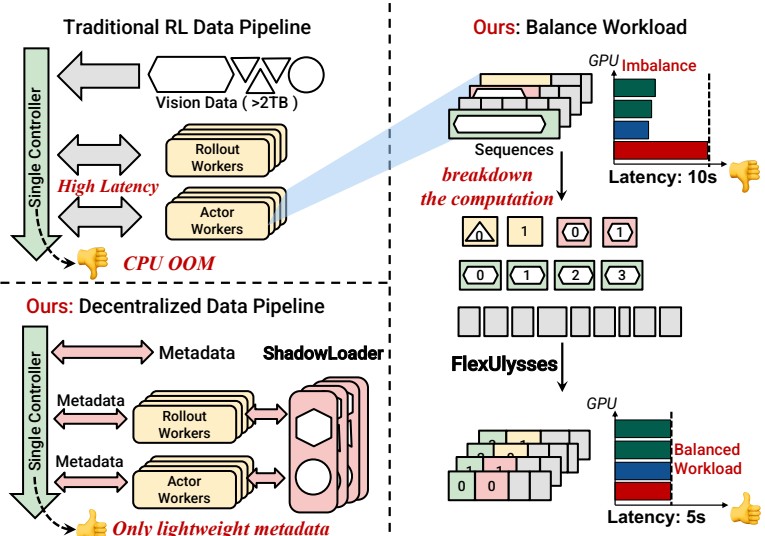

Figure 1: **FlexRL Overview**. (Left) Traditional VLM RL pipelines centralize vision-data fetching and preprocessing with a single controller, leading to high latency and CPU out-of-memory (OOM). ShadowLoader decentralizes the pipeline by parallelizing heavy I/O and preprocessing across rollout/actor workers while the controller exchanges only lightweight metadata. (Right) Highly variable multimodal sequence lengths cause severe cross-GPU imbalance when sequences are scheduled as indivisible units. FlexUlysses shards sequences into fine-grained chunks to balance computation for both rollout inference and model updates, reducing step latency ($10\,\text{s} \rightarrow 5\,\text{s}$ in the example).

parallelism methods over DP instances and gradient steps, which require large batch sizes and gradient accumulation to be effective. This assumption breaks down in the context of RL, which often utilizes small batch sizes to maximize the utility of dynamically generated samples and maintain training stability. These methods are thus inefficient for the dynamic and iterative nature of the RL rollout-update loop.

We present FlexRL, a system designed to provide comprehensive, end-to-end optimization for the VLM RL pipeline. FlexRL deconstructs the performance bottlenecks in each stage of the RL process and introduces targeted solutions. For the data-loading bottleneck, we design ShadowLoader, a data pipeline that parallelizes expensive data fetching and preprocessing tasks across all worker nodes and handles only lightweight sample metadata on the single controller. This decentralizes the workload, eliminating the master-node bottleneck and significantly accelerating data throughput for large media files. For inference and update imbalance, we propose FlexUlysses, a sequence-sharding mechanism. Instead of treating sequences as indivisible units, we partition them into fine-grained chunks. This allows FlexRL to balance the load at a sub-sequence level, effectively mitigating the imbalance caused by extreme length variations during both the inference and training steps. This is complemented by a specialized balancing strategy for the vision tower. FlexRL incorporates an efficient decision algorithm to determine the optimal sharding strategy for each sequence and a dynamic execution engine to orchestrate complex computation and communication patterns, maximizing hardware utilization.

We implement FlexRL on top of the veRL framework and conduct extensive experiments on two 128-GPU clusters (H800 and H200). Our updated evaluation demonstrates that by optimizing the entire RL workflow, FlexRL achieves up to an $8.47\times$ improvement in end-to-end throughput over the baseline. End-to-end ablations confirm that ShadowLoader and FlexUlysses provide complementary gains when combined.

In summary, our contributions are:

- We provide an end-to-end diagnosis of VLM RL training and identify two primary bottlenecks: centralized multimodal data loading and severe cross-GPU workload imbalance.

- We propose **ShadowLoader**, a distributed, metadata-driven data pipeline that parallelizes vision-data fetching and preprocessing across worker nodes and enables metadata-only scheduling on the controller. With prefetching and asynchronous materialization, ShadowLoader nearly breaks the data bottleneck by removing data processing and loading latency from the training critical path.

- We propose **FlexUlysses**, an adaptive sub-sequence sharding mechanism with a cost-aware decision model and a deadlock-free dynamic execution engine that overlaps communication and computation, achieving fine-grained load balancing for both rollout inference and model updates under extreme multimodal length skew.

- Our extensive evaluation on large-scale GPU clusters demonstrates that FlexRL significantly outperforms existing baselines, achieving up to an 8.47× improvement in end-to-end throughput.

## 2 LOAD BALANCING CHALLENGES IN VLM RL TRAINING

### 2.1 DATA PREPARATION BOTTLENECK

Unlike text-only RL, multimodal RL must repeatedly fetch and pre-process large visual payloads (e.g., decoding video clips and sampling frames) on the critical path. Many RL frameworks, such as VeRL (Sheng et al., 2025), adopt a hybrid-controller architecture that centralizes these CPU- and I/O-heavy operations on a single master node, which then dispatches materialized tensors to workers. As the global batch size grows, the master's preprocessing time and host-memory footprint scale with the amount of visual data, quickly turning it into the pipeline straggler and frequently triggering CPU out-of-memory (OOM) errors. Figure 2 illustrates this effect: in VeRL, data loading accounts for 57.1% of the step time, exceeding rollout (18.0%), inference (15.5%), and update (9.3%) combined. This motivates a distributed, metadata-driven data pipeline (ShadowLoader; Sec. 4) that parallelizes preprocessing and hides data movement behind GPU computation.

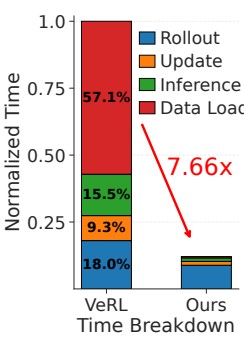

Figure 2: Iteration time breakdown of GRPO.

### 2.2 COMPUTE AND MEMORY IMBALANCE IN MODEL EXECUTION

Beyond data loading, the execution phase suffers from severe compute and memory imbalances. First, attention computation scales quadratically with sequence length ($O(L^2)$), while activation memory scales linearly ($O(L)$). A placement strategy that equalizes memory across GPUs can still result in severe compute imbalance, and vice versa. Second, multimodality introduces heterogeneous costs that are poorly correlated with text length. The computational cost of the vision tower depends heavily on the number of images or video frames. Mixing video-heavy and image-heavy samples within the same batch creates massive variance in per-sample compute and memory requirements, making length-only bucketing objectives ineffective.

### 2.3 WHY EXISTING LOAD BALANCING FALLS SHORT IN VLM RL

**Fixed-parallelism bucketing/packing.** Classic bucketing methods (Team et al., 2025b;c; Wang et al., 2025b; Team et al., 2025d) sort sequences by length and pack them into per-GPU buckets under a fixed parallelism configuration. While simple, this strategy fails when a single long outlier dominates the step. For instance, as shown in Figure 3(Middle), with sequences of varying lengths (e.g., 32K, 16K, and 4K) on four GPUs, no matter how sequences are bucketed, the GPU processing the 32K sample becomes the bottleneck, leading to unavoidable slowdowns. This issue persists in 3D/4D parallelism settings that assume homogeneous configurations across DP ranks.

**Heterogeneous DP across buckets.** Methods such as FlexSP (Wang et al., 2025c), HotSPa (Ge et al., 2024), Hydraulis (Li et al., 2024), and ByteScale (Ge et al., 2025) assign different parallelism configurations to different buckets. However, they rely on large batch sizes and gradient accumulation to create sufficient scheduling space for sequence reordering. In RL, frequent rollout-update alternations and small batch sizes make re-sharding, extra synchronization, and optimizer-

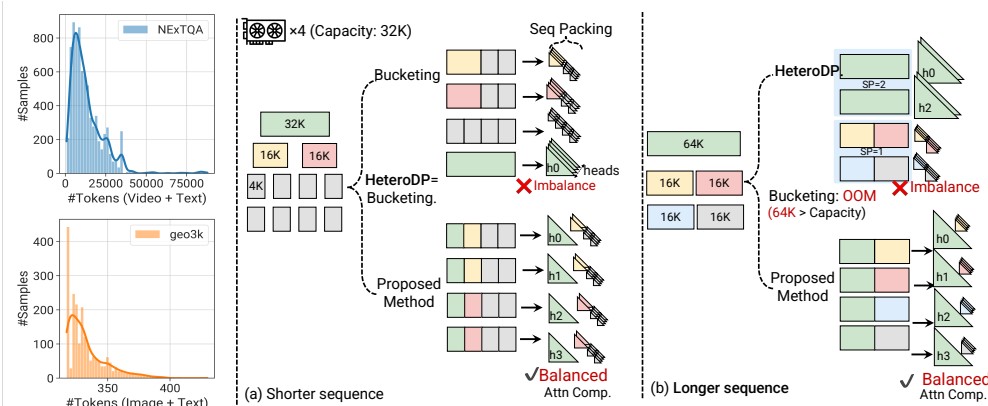

Figure 3: Motivation and idea of FlexUlysses. **Left**: Distribution of token counts in typical video-text (NExTQA (Xiao et al., 2021)) and image-text (Geo3K(Lu et al., 2021)) datasets, showing extreme variation in sequence lengths. **Middle**: An imbalance with shorter sequences, where conventional bucketing leads to load imbalance. **Right**: For longer sequences, bucketing can cause out-of-memory (OOM) errors and further imbalance, while our approach enables fine-grained sharding and balanced attention computation across GPUs.

state movement prohibitively expensive. Furthermore, when sequence parallelism (SP) is enabled, short sequences suffer from redundant communication overhead, and assigning a single SP degree per bucket fails to accommodate mixed-length samples.

# 3 FLEXULYSSES

## 3.1 PRELIMINARY: ULYSSES SEQUENCE PARALLELISM

In the Ulysses approach, a sequence $\mathbf{x}$ of length $L$ is split along the sequence dimension across $N$ devices. Each device $i \in \{0, \ldots, N-1\}$ receives an equal-sized chunk $\mathbf{x}_i$ of length $L/N$. During the forward pass of a layer, each device computes its local Query ($\mathbf{Q}_i$), Key ($\mathbf{K}_i$), and Value ($\mathbf{V}_i$) tensors from its chunk $\mathbf{x}_i$. To compute the full attention scores, the $\mathbf{K}_i$ and $\mathbf{V}_i$ tensors must be shared among all $N$ devices. This is achieved via an `all-to-all` communication operation. After the `all-to-all`, each device possesses the complete Key and Value tensors for a subset of attention heads, allowing it to compute its shard of the attention output. Another `all-to-all` operation is then performed to gather the output, which is passed to subsequent layers.

## 3.2 SEQUENCE SHARDING FOR LOAD BALANCING

Bucketing fails in scenarios like Figure 3(Middle), where the *maximum* sequence length is not large enough to require sequence parallelism (SP) for memory, yet the batch exhibits severe length skew. In this regime, the iteration time is dominated by the single longest sequence: regardless of how we bucket or pack samples, the DP rank that receives this outlier becomes the straggler.

We note that Ulysses sequence parallelism implements an equal-cost partition of a sequence. In the MLP module, each GPU receives a shard of the same length, resulting in identical computation and memory footprints. In the attention module, Ulysses computes attention at the head level, and the computation for each head is identical. Therefore, using Ulysses naturally equalizes the computation and memory footprint across GPUs, and it is applicable to any attention pattern. Consequently, applying Ulysses to packed sequences can also achieve perfect load balancing, as sequence packing is essentially a specific attention pattern (concatenating multiple causal attentions). This differs from RingAttention, which, despite partitioning the sequence, requires meticulously designed partitioning decisions and scheduling for different attention patterns. This inspired us to repurpose this mechanism as a load-balancing primitive: using Ulysses chunks as scheduling units and allocating

them across different devices to balance the workload on each GPU, even when original sequence lengths are highly variable.

**Strawman Solution: Greedy Sharding.** The most straightforward approach is to pack all sequences in the batch and then apply Ulysses, which would force load balancing across all GPUs. However, it is impractical for two reasons. First, Ulysses' scalability is capped by the number of attention heads, limiting the effective degree of parallelism. Second, it incurs substantial communication overhead. While the attention computation for packed sequences scales roughly as $O(\sum_i L_i^2)$, the `all-to-all` communication volume grows with the total token count, increasing the communication-to-computation ratio and leading to a higher GPU idle ratio.

**Our Solution: FlexUlysses.** FlexUlysses addresses this by *adaptively* sharding sequences: instead of sharding every sequence into the same (maximal) number of chunks, it assigns each sequence a sharding degree based on its length and then buckets sequences accordingly. This brings two advantages. First, most sequences do not need to be sharded or only require light sharding to achieve near-optimal load balancing, without incurring communication overhead for the entire batch. Second, since sequences with different SP degrees are processed by different device groups, the `all-to-all` communication and attention computation are independent across sequences (and across different device groups), which enables pipelining and overlapping communication with computation to reduce idle time (bubbles).

**Key Challenges.** However, this hybrid approach introduces two significant implementation challenges. First, the search space is prohibitively large. Finding an optimal configuration requires solving a two-level combinatorial problem: (a) *Sharding Degree Selection*, which chooses a sharding degree for each of the sequences, yielding a search space that grows exponentially with batch size; (b) *Device Group Placement*, which assigns a concrete GPU group to each sharded sequence. The latter is a constrained optimization problem akin to an NP-hard bin-packing problem, because placements across sequences are coupled and must jointly satisfy per-GPU resource limits.

Second, the resulting configuration creates a complex co-scheduling problem for computation and communication, which breaks the conventional SPMD paradigm. Each sequence is partitioned into multiple chunks and mapped to different device groups. Chunks on the same GPU may belong to different sequences and thus require different `all-to-all` groups for communication. These groups may overlap, and an incorrect execution order can lead to deadlocks. Moreover, since `all-to-all` operations are collective and require synchronization, a naive implementation that serializes the communication for each group would introduce significant GPU idle time (bubbles), diminishing the benefits of FlexUlysses. Scheduling the computation and communication of these chunks to maximize GPU utilization is a non-trivial problem. Therefore, a sophisticated scheduling mechanism is required to manage these diverse computation and communication patterns efficiently.

## 3.3 Planning and Scheduling for Load Balancing

We design a two-level planning and scheduling mechanism to address these challenges. It first determines per-sequence sharding degrees and device groups. Then it orchestrates the Ulysses execution order for the assigned sharded sequences on each GPU.

**Shard Planning.** Given a global batch, the single controller collects lightweight metadata and outputs a sharding plan. For each sequence $i$ with length $h_i$, we choose a sharding degree $p_i \in \{1, 2, 4, \ldots, p_{\max}\}$ and assign it to a device group $G_i$ with $|G_i| = p_i$. We pre-instantiate a *hierarchical* set of candidate device groups for all admissible degrees: for each degree $p$, the groups form a partition of GPUs (disjoint and covering), and partitions are nested across degrees (a $p$-group is the union of two $p/2$-groups). For example, on 8 GPUs, the admissible groups include [0-7] for $p$=8, [0-3] and [4-7] for $p$=4, and [0, 1], [2, 3], [4, 5], [6, 7] for $p$=2. This hierarchy ensures that any two groups are either disjoint or one contains the other, which enables a simple deadlock-free schedule. In practice, we decouple grouping from placement by first instantiating all admissible groups, then assigning sequences to groups using a lightweight greedy heuristic (Appendix A).

**Highest-Sharding-First Execution.** Collective primitives are order-sensitive: for a given communicator, all ranks must issue collectives in the same order; otherwise, execution can hang. With multiple sharding degrees, a GPU can participate in multiple (nested) device groups, and a naive, per-rank execution order can deadlock even when groups are hierarchical. For example, on 4 GPUs,

---

**Algorithm 1** FlexUlysses Forward Attention Execution.

---

**Require:** packed sequence groups $\mathcal{Q}, \{(p(SG), G(SG))\}_{SG \in \mathcal{Q}}$
**Require:** global HSF-ordered list $\mathcal{O}$ over $\{SG \in \mathcal{Q} : p(SG) > 1\}$
    **Worker rank** $k$ **(execution)**
1:  $\mathcal{O}_k \leftarrow [SG \in \mathcal{O} \mid k \in G(SG)]$                *▷ ordered subsequence*
2:  $\mathcal{U}_k \leftarrow [SG \in \mathcal{Q} \mid k \in G(SG) \wedge p(SG) = 1]$       *▷ no collectives*
3: **if** $\mathcal{O}_k \neq \emptyset$ **then**
4:     launch `all2all1`$(\mathcal{O}_k[1])$ asynchronously         *▷ comm stream*
5: RUNUNSHARDED$(\mathcal{U}_k)$                      *▷ hide `all2all1`*
6: **for** $t = 1$ **to** $|\mathcal{O}_k|$ **do**
7:     $SG \leftarrow \mathcal{O}_k[t]$
8:     wait `all2all1`$(SG)$
9:     **if** $t < |\mathcal{O}_k|$ **then**
10:        launch `all2all1`$(\mathcal{O}_k[t+1])$ asynchronously     *▷ issued at compute start*
11:     compute attention$(SG)$
12:     launch `all2all2`$(SG)$ asynchronously           *▷ comm stream*

---

consider a 4-way group $G_4 = \{0, 1, 2, 3\}$ and a nested 2-way group $G_2 = \{0, 1\}$. If rank 0 enters an `all2all` on $G_2$ while ranks 1–3 enter an `all2all` on $G_4$, then rank 0 waits for rank 1 to join $G_2$, while rank 1 waits for rank 0 to join $G_4$, forming a cyclic wait.

FlexUlysses avoids deadlocks by combining hierarchical device groups with a highest-sharding-first (HSF) schedule. HSF then enforces a consistent global order across all ranks: each GPU always executes collectives for larger degrees before smaller ones (e.g., $p=8 \rightarrow 4 \rightarrow 2$), and within the same $(p, G)$, it follows a consistent order so the collective call sequence matches across ranks. As a result, a GPU never interleaves collectives from multiple device groups at the same time. With HSF, any rank that belongs to both $G_4$ and $G_2$ must enter $G_4$ collectives before $G_2$ collectives, preventing the cyclic wait.

**Vision Tower Balancing.** Unlike the LLM backbone, the vision tower is naturally parallelizable across images/frames. Vision encoders typically process images independently by stacking them along the batch dimension, and video frames are sampled and processed with intra-frame attention, making computation independent across frames. As a result, vision-tower compute and memory scale near-linearly with the number of images or frames. We leverage this property by distributing images and video frames evenly across GPUs to balance both compute and memory loads.

### 3.4 DYNAMIC PACKING AND OVERLAPPING

**Dynamic Shard Packing.** Executing HSF at single-sequence granularity would repeatedly launch small attention kernels and two `all2all` collectives per sequence, which can incur non-trivial launch overhead and leave little room to hide communication. We therefore pack sequences assigned to the same device group into *sequence groups* and run Ulysses on each group as a unit. Concretely, within each $(p, G)$ we greedily concatenate *short* sequences (e.g., $h_i \leq h_{\text{pack}}$) until the packed token count reaches a cap $H_{\text{pack}}$, while keeping long sequences as singleton groups. This bounded packing trades off between overhead amortization and scheduling flexibility: larger groups reduce kernel/collective launch frequency and create longer compute windows to cover `all2all`, while the caps prevent a pack from becoming a new straggler and preserve enough groups for pipelining.

**Overlapping.** For a single sequence group, Ulysses induces a strict dependency `all2all1` → `attn comp` → `all2all2`. Different sequence groups are independent, so we pipeline them to overlap communication with computation, as illustrated in Fig. 4. When processing sequence groups under HSF, once $SG_j$ enters attention computation, we immediately launch `all2all1` of the next group $SG_{j+1}$ on the communication stream.

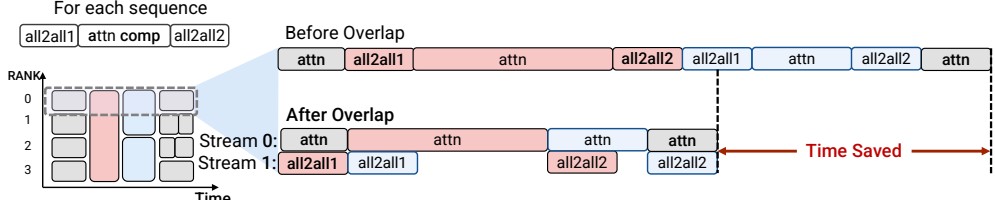

Figure 4: Sliced sequence computation pattern of FlexUlysses and communication overlapping strategies.

# 4    SHADOWLOADER FOR THE DATA-LOADING BOTTLENECK

To overcome the data loading bottleneck in the hybrid-controller architecture, we design **ShadowLoader**, a distributed, metadata-driven data-loading system. ShadowLoader consists of four key components: (1) **Proxy Dataloader**, which resides on the single controller to manage lightweight metadata and dispatch preprocessing tasks; (2) **Local Preprocessor**, deployed on each worker node to independently handle heavy preprocessing tasks such as video decoding and frame sampling; (3) **MetaStore**, a registry that maintains the mapping between sample metadata and its physical storage location; and (4) **Materializer**, instantiated on each worker to fetch the materialized visual tensors from the corresponding Local Preprocessor based on metadata.

On the single controller, the Proxy Dataloader replaces the standard dataloader. For multimodal data, it only retains lightweight metadata, such as video length, expected frame count, image size, resolution, and file path. To seamlessly integrate with the existing RL training pipeline without loading actual pixel values into the controller's memory, the Proxy Dataloader uses a `FakeTensor` as a placeholder for visual data. The `FakeTensor` contains shape information (e.g., `pixel_values`) but no actual data, and it overrides the `shard` operation to support subsequent Ulysses partitioning decisions. During training, these `FakeTensors` are distributed across different workers according to the batching and load-balancing strategies.

**Overall Workflow.** The end-to-end data loading process proceeds as follows. First, the Proxy Dataloader assigns a unique ID to each data sample and dispatches it to a Local Preprocessor using a simple load-balancing algorithm (e.g., routing to the node with the least load). Upon receiving the task, the Local Preprocessor registers the mapping between the data's metadata and its storage location in the MetaStore. It then executes the preprocessing and caches the materialized visual tensors in its local host memory. Meanwhile, the single controller schedules the RL training loop using the `FakeTensors`. When a worker receives a `FakeTensor` and requires the actual visual data for computation, it invokes its local Materializer. The Materializer queries the MetaStore using the metadata to locate the target preprocessor and fetches the materialized tensors directly from the corresponding Local Preprocessor.

To further remove data processing from the critical path and minimize overhead, we introduce two additional optimizations to reduce transfer volume and hide latency:

**Prefetching and Asynchronous Materialization.** ShadowLoader overlaps data movement with training by prefetching the metadata for the next step and materializing visual tensors asynchronously on the workers. By launching non-blocking materialization, CPU-side fetching, decoding, and network transfers are completely hidden behind ongoing GPU computation, effectively eliminating step-level bubbles caused by large visual payloads.

**FlexUlysses-aware Loading.** We further co-design ShadowLoader with FlexUlysses by making sharding decisions early and loading only the required data slices. Because the `FakeTensor` placeholders carry slice-aware metadata, workers can fetch precisely what they need—such as specific frame ranges for videos or required image tensors for multi-image inputs. Aligning this slice-aware materialization with FlexUlysses partition decisions significantly reduces inter-node transfer volume and improves end-to-end throughput.

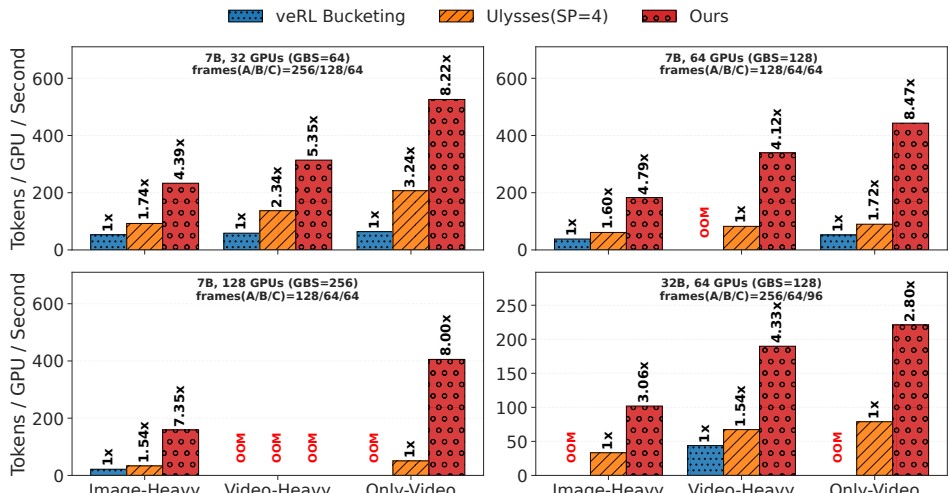

Figure 5: End-to-end throughput (tokens/s/GPU) across three dataset settings (Image-Heavy/Video-Heavy/Only-Video) under different model scales and cluster sizes. Geo3K, NExTQA, and LongVILA-Reason mixtures with sampling weights 5:2:1, 1:2:5, and only LongVILA-Reason, respectively. Speedup labels are computed against veRL Bucketing when available, otherwise against Ulysses (SP=4). Red **OOM** markers CPU or GPU out-of-memory.

## 5 EVALUATION

### 5.1 EVALUATION SETUP

**Testbed.** We evaluate FlexRL on two 128-GPU clusters (each 16×8): one with NVIDIA H800 GPUs and one with NVIDIA H200 GPUs. Intra-node GPU interconnect bandwidth is 400GB/s (H800) and 900GB/s (H200) via NVLink/NVSwitch. Inter-node connectivity uses RoCEv2 RDMA, with aggregate per-node bandwidth of 1600Gb/s (H800 cluster) and 3200Gb/s (H200 cluster). Unless otherwise stated, end-to-end comparisons are reported on the H800 cluster.

**Models and Datasets.** We evaluate FlexRL on MiMo-VL-7B-RL (Team et al., 2025a) and Qwen2.5-VL-32B (Bai et al., 2025). We benchmark on a mixture of image-text (Geo3K (Lu et al., 2021)), short-video (NExTQA (Xiao et al., 2021)), and long-video (LongVILA-Reason (Chen et al., 2025b)) datasets under different sampling weights: (1) Image-Heavy: 5:2:1; (2) Video-Heavy: 1:2:5; (3) Only-Video: LongVILA-Reason only. We use the GRPO for all experiments, with a maximum response length of 1024 tokens. For the 7B model, we use TP=4 for rollout and FSDP=16 for model forward/update. For the 32B model, we use TP=16 for rollout and FSDP=64 for model forward/update. We set the maximum sharding budget $p_{max}$ to 8 for all runs unless otherwise specified.

### 5.2 MAIN RESULTS

**End-to-end Results.** We compare FlexRL with two baselines: (1) **veRL+Bucketing**, the original veRL system with bucketing; and (2) **Ulysses-SP**, conventional sequence parallelism with a fixed sharding degree. Figure 5 reports end-to-end throughput (tokens/GPU/s) for GRPO under different dataset sampling weights, global batch sizes, and maximum numbers of video frames. Overall, FlexRLconsistently outperforms both veRL+Bucketing and fixed-degree Ulysses-SP. In particular, FlexRLimproves end-to-end throughput by up to $8.47\times$ over the baselines. In image-heavy settings, FlexRLachieves up to a $7.35\times$ speedup, while in video-heavy settings the speedup reaches up to $5.35\times$. In the only-video setting, where sequence lengths are longest and most heterogeneous, FlexRLachieves the largest gain of up to $8.47\times$. The improvements are more pronounced for larger models and video-heavy datasets, where load imbalance is more severe. Moreover, throughput scales well with the video volume, demonstrating FlexRL's effectiveness in handling long sequences. In contrast, the veRL baseline suffers from severe slowdowns and out-of-memory (OOM) issues due to ViT-induced load imbalance as the number of video samples increases. Ulysses-SP mitigates the

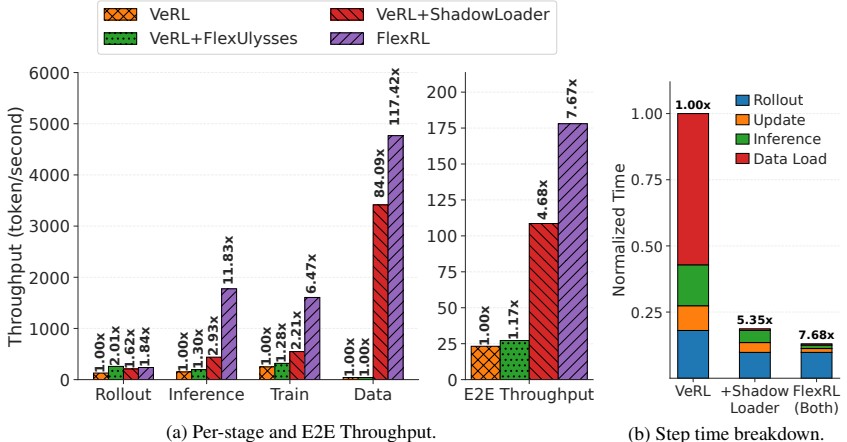

(a) Per-stage and E2E Throughput.

(b) Step time breakdown.

Figure 6: Ablation results on 128 H800 GPUs with MiMo-VL-7B-RL. (a) Throughput (tokens/s) for each RL stage and the end-to-end throughput, along with the speedup against VeRL. (b) Normalized iteration time breakdown.

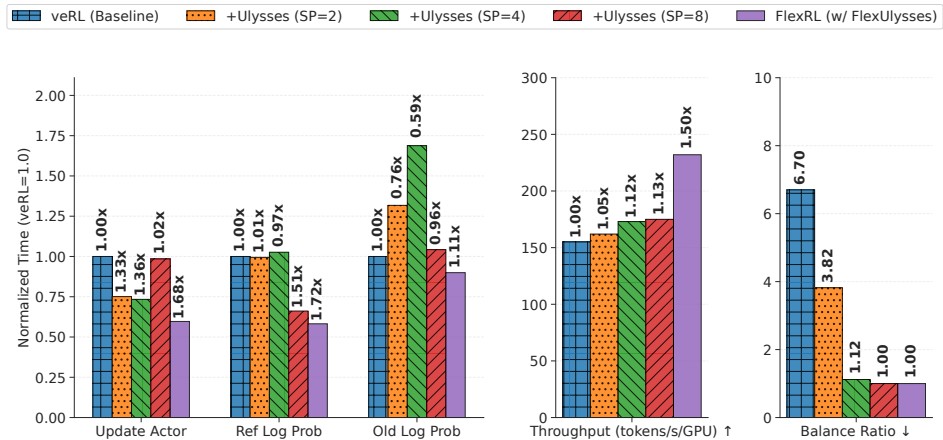

Figure 7: Comparison with Ulysses-SP. Per-stage normalized step time with the speedup (Left). End-to-end throughput (Middle). Balance ratio calculated using the attention flops (Right).

veRL baseline by balancing attention computation; however, its fixed sharding degree incurs substantial communication overhead, and its performance gain diminishes as the batch size increases.

**Comparison with Ulysses-SP.** We evaluate our method and Ulysses-SP method using the image-heavy dataset and 7B model on 32 GPUs. We enable ShadowLoader for all methods to eliminate the influence of dataloading, and compare fixed-degree Ulysses-SP (SP=$\{2, 4, 8\}$) against ours. We set veRL+Bucketing as the baseline. The attention *balance ratio* equals the maximum per-GPU attention FLOPs / mean FLOPs. As shown in Figure 7, when using Ulysses-SP, increasing the SP degree can also achieve load balance. However, the throughput gain of Ulysses-SP is limited due to the all2all communication dominating overhead. In contrast, FlexUlysses succeeds to reaches a balance ratio of 1.0 while improving throughput by 1.50×. Figure 7 further breaks down the per-stage step time. The results show that our method achieves the highest speedup across all stages, demonstrating the effectiveness of our fine-grained sharding strategy in improving load balance and reducing communication overhead.

## 5.3 ABLATION STUDY.

We separately evaluate the contributions of ShadowLoader and FlexUlysses under the image-heavy setting using the 7B model. Figure 6 reports both the throughput breakdown and the iteration time

breakdown. Figure 6a shows that ShadowLoader primarily removes the data bottleneck, improving data-loading throughput by $84.09\times$ and end-to-end throughput by $4.68\times$. Figure 6b further confirms that this translates into a $5.35\times$ reduction in overall step time by largely eliminating the data-loading slice. In contrast, FlexUlysses mainly improves compute-heavy stages (e.g., $2.01\times$ rollout and $1.28\times$ training throughput) but yields limited end-to-end gains ($1.17\times$) when data loading remains dominant. Without ShadowLoader, the single controller has to send the data to all workers, incurring high data transfer latency. Combining both components yields complementary benefits: after ShadowLoader makes the workload compute-bound, FlexUlysses' fine-grained sharding translates into large gains in inference ($11.83\times$) and training ($6.47\times$), reaching $117.42\times$ data-loading throughput, $7.67\times$ end-to-end throughput, and a $7.68\times$ reduction in step time. The detailed per-stage runtimes and throughput are shown in Table 1.

Table 1: Detailed step time breakdown of ablation study of FlexRL components on 128 H800 GPUs with MiMo-VL-7B-RL.

| Method | Update Actor (s) | RefLogProb (s) | OldLogProb (s) | Data Loading (s) | Generation (s) | Step Time (s) |
|---|---|---|---|---|---|---|
| veRL | 61.3 (1.00x) | 51.3 (1.00x) | 50.2 (1.00x) | 375.0 (1.00x) | 118.4 (1.00x) | 656.2 (1.00x) |
| + ShadowLoader | 24.3 (2.52x) | 5.0 (10.26x) | 25.3 (1.99x) | 3.9 (96.15x) | 64.1 (1.85x) | 122.6 (5.35x) |
| + FlexUlysses | 47.8 (1.28x) | 50.1 (1.02x) | 27.8 (1.81x) | 375.0 (1.00x) | 58.9 (2.01x) | 559.6 (1.17x) |
| **FlexRL (Both)** | **9.5 (6.45x)** | **3.7 (13.86x)** | **4.9 (10.24x)** | **3.2 (117.19x)** | **64.4 (1.84x)** | **85.7 (7.66x)** |

## 6 RELATED WORK

**RL Training Frameworks.** Reinforcement learning has become a central paradigm for aligning and enhancing large language models (LLMs). Recent frameworks such as VeRL (Sheng et al., 2025), siiRL (Wang et al., 2025e), AReal (Fu et al., 2025), StreamRL (Zhong et al., 2025), MiroRL (Team & Team, 2025), and ROLL (Wang et al., 2025a) provide system-level support for distributed RL training. These frameworks focus on issues such as asynchronous rollout-update decoupling, scalable data pipelines, and integration with model-parallel training backends.

**VLM Training Frameworks.** The emergence of multimodal LLMs has motivated training frameworks such as DistTrain (Zhang et al., 2025), DistMM (Huang et al., 2024), LongVILA (Chen et al., 2025b), and VeOmni (Ma et al., 2025). These systems propose optimizations for heterogeneous architectures combining vision towers and language backbones. For instance, they disaggregate model components, employ hybrid parallelism, or develop scheduling algorithms to reduce communication overheads. Despite these advances, existing VLM frameworks primarily target pretraining or supervised fine-tuning. They do not explicitly address the unique challenges of RL training, such as small batch sizes, dynamically generated trajectories, and highly variable sequence lengths across modalities.

**Load Balancing for Large Model Training.** A line of work focuses on load balancing techniques for efficient large model training. Classic methods rely on sequence bucketing and packing, which sort samples by length and allocate them across GPUs to minimize padding (Team et al., 2025b;c;d; Wang et al., 2025b). Recent works such as FlexSP (Wang et al., 2025c), HotSPa (Ge et al., 2024), Hydraulis (Li et al., 2024), ByteScale (Ge et al., 2025), and WLB-LLM (Wang et al., 2025d) mitigate load imbalance via heterogeneous parallelism and dynamic reconfiguration across DP ranks or gradient steps. Zeppelin (Chen et al., 2025a) addresses training imbalance by jointly optimizing attention-specific sequence partitioning and layout remapping between attention and linear layers.

## 7 CONCLUSION

In this work, we present FlexRL, a holistic system that addresses the unique system-level challenges of reinforcement learning for large Vision-Language Models. By systematically analyzing the bottlenecks across the RL pipeline, we identify critical inefficiencies in both data loading and workload balancing that hinder scalability and hardware utilization. FlexRL introduces ShadowLoader to eliminate I/O and memory bottlenecks on the single controller, and FlexUlysses to achieve fine-grained, shard-level load balancing across GPUs. Our efficient scheduling algorithm and dynamic execution engine further maximize overlap between computation and communication. Combining both components achieves high throughput even under extreme data heterogeneity.

ACKNOWLEDGEMENTS

We thank the anonymous reviewers and the area chair (AC) for their insightful comments. This work is supported by the Shanghai Municipal Science and Technology Major Project and the Shanghai Artificial Intelligence Laboratory.

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

## A  SEQUENCE PLACEMENT ALGORITHM

**Problem formulation.**  We formalize planning over $D$ devices for a global batch of $N$ sequences. Let each sequence $i \in \{1, \ldots, N\}$ have length $h_i$. We choose a sharding degree $p_i \in \{1, 2, 4, \ldots, p_{\max}\}$ and an assignment $y_{i,d} \in \{0, 1\}$ indicating whether device $d$ holds a shard of sequence $i$, such that $\sum_{d=1}^{D} y_{i,d} = p_i$. For executability, we restrict sharded sequences $(p_i > 1)$ to a hierarchical family of admissible device groups: for each degree $p$, the groups form a partition of GPUs (disjoint and covering), and partitions are nested across degrees.

**Cost model.**  On each device $d$, we define (i) **memory load** $M_d = \sum_{i=1}^{N} (h_i/p_i) \, y_{i,d}$ with a hard threshold $M_d \leq M_{\text{th}} = \alpha M^\star$ (we use $\alpha = 1.1$), where $M^\star$ is the average per-device token count; and (ii) **compute load** $C_d \propto \sum_{i=1}^{N} (h_i^2/p_i) \, y_{i,d}$ as an attention-cost proxy. Sharding also introduces communication overhead; following the standard SeqAlltoAll model, the per-device traffic volume scales as $\text{vol}(h, p) \propto h \cdot (p-1)/p^2$, optionally weighted by intra-/inter-node bandwidth. In Algorithm 2, we use $\text{SplitCost}(h, p)$ as a practical proxy for this overhead.

**Objective.**  We solve a lexicographic optimization:

$$\min_{\{p_i, y_{i,d}\}} \left( \max_d C_d, \ \sum_{i=1}^{N} \text{Comm}(h_i, p_i) \right), \tag{1}$$

where we primarily minimize the maximum compute imbalance and use the communication cost to break ties.

**FlexUlysses 3-Stage Scheduling Heuristic.**  The detailed sequence placement algorithm is formalized in Algorithm 2. Our algorithm incorporates a threshold multiplier $(1.1\times)$ for memory constraint $M_{\text{th}}$ and searches over power-of-two candidate degrees.

---

**Algorithm 2** FlexUlysses Placement Algorithm

---

**Require:** sequence heights $\mathcal{H} = \{h_i\}_{i=1}^{N}$, number of devices $D$
**Ensure:** placement $\mathcal{Y} = \{(i, d, \hat{h}_{i,d}, p_i)\}$ and per-item split degree $\mathcal{P} = \{p_i\}$
1: $M^\star \leftarrow \frac{1}{D} \sum_{i=1}^{N} h_i, \quad M_{\text{th}} \leftarrow \alpha M^\star$          $\triangleright \alpha = 1.1$
2: $p_{\max} \leftarrow 2^{\lfloor \log_2 D \rfloor}, \quad \mathcal{P}_{\text{cand}} \leftarrow \{2^k : 0 \leq k \leq \log_2 p_{\max}\}$
3: $(\mathcal{Y}, \mathbf{m}) \leftarrow \text{BASELINESPLITPLACE}(\mathcal{H}, D, M^\star, M_{\text{th}}, \mathcal{P}_{\text{cand}})$
4: $\mathbf{s} \leftarrow$ item states induced by $\mathcal{Y}$      $\triangleright \mathbf{s}[i].p$=*current split degree*, $\mathbf{s}[i].G$=*device group*
5: $\mathcal{L} \leftarrow \{2^k : 1 \leq k \leq \log_2 p_{\max}\}$ in decreasing order
6: **for all** $p \in \mathcal{L}$ **do**
7:     **while** $\exists$ merge candidate at split level $p$ **do**
8:         $\mathcal{C}_p \leftarrow \text{FINDMERGECANDIDATES}(\mathcal{Y}, \mathbf{s}, p, M_{\text{th}}, m)$        $\triangleright m = $
     $DEFAULT\_M\_CANDIDATES$
9:         **if** $\mathcal{C}_p = \emptyset$ **then**
10:            **break**
11:         $c^\star \leftarrow \arg\min_{c \in \mathcal{C}_p} \Delta\text{Cost}(c)$
12:         $\text{EXECUTEMERGE}(\mathcal{Y}, \mathbf{s}, c^\star)$
13: $\text{SHORTITEMREORDER}(\mathcal{Y}, \mathbf{s}, \mathcal{H}, M_{\text{th}})$
14: **return** $\mathcal{Y}, \mathcal{P}$

---

**Split cost.**

$$\text{SplitCost}(h, p) = \begin{cases} 0, & p \leq 1 \\ \frac{h(p-1)}{p^2}, & 2 \leq p \leq \tau \\ \gamma \cdot \frac{h(p-1)}{p^2}, & p > \tau \end{cases} \quad \text{with } \tau = 8, \ \gamma = 16.$$

$$\text{TotalCost}(\mathcal{P}) = \sum_{i=1}^{N} \text{SplitCost}(h_i, p_i).$$

---

**Algorithm 3** FlexUlysses Sub-procedures

---

1: **procedure** BASELINESPLITPLACE($\{h_i\}_{i=1}^N, D, M^\star, M_{\text{th}}, \mathcal{P}_{\text{cand}}$)
2:     $\mathbf{m} \leftarrow \mathbf{0}, \quad \mathbf{a} \leftarrow 1.1 M^\star \cdot \mathbf{1}$         ▷*available memory proxy*
3:     sort items by $h_i$ descending
4:     **for** $i = 1$ **to** $N$ **do**
5:         $p \leftarrow$ GETSPSIZE($h_i, \mathcal{P}_{\text{cand}}$)         ▷*power-of-two*
6:         **while** $p \leq p_{\max}$ **and** VALIDGROUPEXISTS($p, i, \mathbf{m}, M_{\text{th}}$) = **false do**
7:             $p \leftarrow 2p$
8:         **if** VALIDGROUPEXISTS($p, i, \mathbf{m}, M_{\text{th}}$) = **false then**
9:             $p \leftarrow p_{\max}$
10:         $G \leftarrow \arg\max_{G:|G|=p, \; \forall d \in G: m_d + \frac{h_i}{p} \leq M_{\text{th}}} \sum_{d \in G} a_d$
11:         **for all** $d \in G$ **do**
12:             add tuple $(i, d, \hat{h}_{i,d} = \frac{h_i}{p}, p)$ to $\mathcal{Y}$
13:             $m_d \leftarrow m_d + \frac{h_i}{p}$
14:             $a_d \leftarrow a_d - \frac{h_i}{p}$         ▷*proxy update*
15:     **return** $\mathcal{Y}, \mathbf{m}$

16: **procedure** FINDMERGECANDIDATES($\mathcal{Y}, \mathbf{s}, p, M_{\text{th}}, m$)
17:     partition devices into consecutive groups $\mathcal{G}_p = \{G \subseteq [D] : |G| = p\}$
18:     $\mathcal{C}_p \leftarrow \emptyset$
19:     **for all** $G \in \mathcal{G}_p$ **do**
20:         $I_G \leftarrow \{i : \mathbf{s}[i].p = p \wedge \mathbf{s}[i].G = G\}$
21:         **if** $|I_G| < 2$ **then**
22:             **continue**
23:         **for** $m' = 2$ **to** $m$ **do**
24:             choose a multiset $S \subseteq I_G$ with $|S| = m'$ (greedy by increasing $h_i$)
25:             $(S_1, S_2) \leftarrow$ TWOWAYPACK($S$)         ▷*subject to balance constraints*
26:             **if** $S_1 \neq \emptyset \wedge S_2 \neq \emptyset \wedge$ MERGEFEASIBLE($G, S_1, S_2, p, M_{\text{th}}$) **then**
27:                 $\mathcal{C}_p \leftarrow \mathcal{C}_p \cup \{c = (G, S_1, S_2, p)\}$
28:     **return** $\mathcal{C}_p$

29: **procedure** EXECUTEMERGE($\mathcal{Y}, \mathbf{s}, c = (G, S_1, S_2, p)$)
30:     $p' \leftarrow p/2, \quad$ split $G$ into halves $G^{(1)}, G^{(2)}$ with $|G^{(1)}| = |G^{(2)}| = p'$
31:     remove all shards of items in $S_1 \cup S_2$ from devices in $G$
32:     **for all** $d \in G^{(1)}$ **do**
33:         **for all** $i \in S_1$ **do** add $(i, d, h_i/p', p')$ to $\mathcal{Y}$
34:     **for all** $d \in G^{(2)}$ **do**
35:         **for all** $i \in S_2$ **do** add $(i, d, h_i/p', p')$ to $\mathcal{Y}$
36:     **for all** $i \in S_1$ **do** $\mathbf{s}[i].p \leftarrow p', \mathbf{s}[i].G \leftarrow G^{(1)}$
37:     **for all** $i \in S_2$ **do** $\mathbf{s}[i].p \leftarrow p', \mathbf{s}[i].G \leftarrow G^{(2)}$

38: **procedure** SHORTITEMREORDER($\mathcal{Y}, \mathbf{s}, \{h_i\}, M_{\text{th}}$)
39:     $S \leftarrow \{i : \mathbf{s}[i].p \leq 2\}$ sorted by decreasing $h_i$
40:     **for all** $i \in S$ **do**
41:         remove all tuples in $\mathcal{Y}$ with item $i$
42:         $\mathbf{s}[i].p \leftarrow 1, \mathbf{s}[i].G \leftarrow \emptyset$
43:     **for all** $i \in S$ **do**
44:         $d^\star \leftarrow \arg\min_{d:m_d + h_i \leq M_{\text{th}}} \left( \sum_{(j,d,\cdot,\cdot) \in \mathcal{Y}} h_j \cdot h_j \right)$
45:         **if** no feasible $d^\star$ **then**
46:             $d^\star \leftarrow \arg\min_d(m_d + h_i)$
47:         add $(i, d^\star, h_i, 1)$ to $\mathcal{Y}$; update $m_{d^\star} \leftarrow m_{d^\star} + h_i$

---

# B   DETAILED END-TO-END RUNTIME BREAKDOWN

**End-to-End Results.**

| CR | Model | GPUs | DS | Method | GBS | Frames | Throughput | DataLoading(s) | Rollout(s) | Old(s) | Ref(s) | Step(s) | Update(s) |
|---|---|---|---|---|---|---|---|---|---|---|---|---|---|
| CR-01 | 7B | 32 | Image-Heavy | ours | 64 | 256 | 233.21 | 15.045 | 31.703 | 6.354 | 3.336 | 46.404 | 4.952 |
| CR-01 | 7B | 32 | Image-Heavy | ulysses_sp | 64 | 256 | 92.50 | 54.873 | 45.607 | 20.866 | 20.337 | 114.868 | 28.002 |
| CR-01 | 7B | 32 | Image-Heavy | verl_bucketing | 64 | 256 | 53.18 | 55.572 | 46.447 | 50.763 | 49.881 | 200.155 | 53.006 |
| CR-02 | 7B | 32 | Video-Heavy | ours | 64 | 128 | 313.99 | 11.838 | 29.641 | 18.318 | 8.108 | 63.934 | 7.777 |
| CR-02 | 7B | 32 | Video-Heavy | ulysses_sp | 64 | 128 | 137.15 | 174.811 | 71.966 | 28.197 | 27.751 | 161.325 | 33.326 |
| CR-02 | 7B | 32 | Video-Heavy | verl_bucketing | 64 | 128 | 58.65 | 179.457 | 71.829 | 97.575 | 95.036 | 363.035 | 98.510 |
| CR-03 | 7B | 32 | Only-Video | ours | 64 | 64 | 525.93 | 6.732 | 19.384 | 3.979 | 3.908 | 32.307 | 4.969 |
| CR-03 | 7B | 32 | Only-Video | ulysses_sp | 64 | 64 | 207.03 | 179.082 | 51.967 | 8.901 | 8.705 | 82.010 | 12.370 |
| CR-03 | 7B | 32 | Only-Video | verl_bucketing | 64 | 64 | 63.96 | 181.610 | 52.853 | 70.287 | 69.811 | 265.406 | 72.386 |
| CR-04 | 32B | 64 | Image-Heavy | ours | 128 | 256 | 101.96 | 17.147 | 61.914 | 15.757 | 13.660 | 104.506 | 13.053 |
| CR-04 | 32B | 64 | Image-Heavy | verl_bucketing | 128 | 128 | 35.86 | 91.377 | 68.888 | 42.439 | 42.309 | 201.830 | 48.099 |
| CR-04 | 32B | 64 | Image-Heavy | verl_bucketing | 128 | 256 | 33.29 | 95.529 | 89.113 | 59.432 | 58.244 | 272.536 | 65.640 |
| CR-05 | 32B | 64 | Video-Heavy | ours | 128 | 64 | 189.79 | 6.184 | 41.596 | 7.063 | 6.606 | 68.579 | 13.197 |
| CR-05 | 32B | 64 | Video-Heavy | ulysses_sp | 128 | 64 | 67.29 | 230.772 | 91.813 | 32.929 | 32.871 | 199.202 | 41.476 |
| CR-05 | 32B | 64 | Video-Heavy | verl_bucketing | 128 | 64 | 43.81 | 236.807 | 94.480 | 68.385 | 67.501 | 305.111 | 74.634 |
| CR-06 | 32B | 64 | Only-Video | ours | 128 | 96 | 221.07 | 9.856 | 63.303 | 17.763 | 9.567 | 113.518 | 22.712 |
| CR-06 | 32B | 64 | Only-Video | ulysses_sp | 128 | 96 | 78.85 | 496.666 | 156.860 | 50.508 | 49.079 | 318.196 | 61.579 |
| CR-06 | 32B | 64 | Only-Video | verl_bucketing | 128 | 64 | 47.21 | 355.949 | 112.491 | 79.096 | 78.757 | 357.820 | 87.350 |
| CR-10 | 7B | 64 | Image-Heavy | ours | 128 | 128 | 183.06 | 10.548 | 19.875 | 9.610 | 6.340 | 39.273 | 3.360 |
| CR-10 | 7B | 64 | Image-Heavy | ulysses_sp | 128 | 128 | 61.09 | 97.044 | 50.916 | 24.396 | 24.136 | 126.604 | 27.065 |
| CR-11 | 7B | 64 | Video-Heavy | ours | 128 | 64 | 339.47 | 6.267 | 21.796 | 7.238 | 4.892 | 38.999 | 4.959 |
| CR-11 | 7B | 64 | Video-Heavy | ulysses_sp | 128 | 128 | 74.93 | 369.454 | 123.124 | 58.135 | 58.773 | 303.776 | 63.573 |
| CR-11 | 7B | 64 | Video-Heavy | ulysses_sp | 128 | 64 | 82.45 | 231.429 | 71.792 | 29.816 | 29.491 | 163.260 | 32.051 |
| CR-12 | 7B | 64 | Only-Video | ours | 128 | 64 | 443.24 | 6.761 | 23.774 | 4.438 | 4.329 | 38.309 | 5.635 |
| CR-12 | 7B | 64 | Only-Video | ulysses_sp | 128 | 64 | 89.94 | 356.238 | 91.013 | 31.649 | 31.392 | 188.718 | 34.536 |
| CR-12 | 7B | 64 | Only-Video | verl_bucketing | 128 | 64 | 52.34 | 349.610 | 92.122 | 77.244 | 75.763 | 324.317 | 79.057 |
| CR-13 | 7B | 128 | Image-Heavy | ours | 256 | 128 | 158.94 | 10.956 | 24.982 | 11.185 | 6.643 | 47.537 | 4.541 |
| CR-13 | 7B | 128 | Image-Heavy | ulysses_sp | 256 | 128 | 33.26 | 182.141 | 74.846 | 49.702 | 48.861 | 225.579 | 51.983 |
| CR-13 | 7B | 128 | Image-Heavy | verl_bucketing | 256 | 128 | 21.63 | 186.269 | 76.725 | 91.113 | 88.703 | 347.586 | 90.853 |
| CR-15 | 7B | 128 | Only-Video | ours | 256 | 64 | 405.44 | 6.973 | 24.783 | 5.229 | 4.924 | 41.885 | 6.692 |
| CR-15 | 7B | 128 | Only-Video | ulysses_sp | 256 | 64 | 50.67 | 694.390 | 149.507 | 61.255 | 60.590 | 335.006 | 63.407 |

Table 2: Detailed E2E experiment results.

**Sequence Packing Micro-Benchmark.** To clarify how the runtime of global-attention-style packing depends on the sequence-length distribution, we micro-benchmark FlashAttention variable-length attention (`flash_attn_var_len`) on a single H800 GPU. We run GQA with $q$ heads = 64, $kv$ heads = 8, and head dim = 128. We keep the total number of tokens fixed at 64K and vary how they are partitioned across sequences (Table 3).

Table 3: `flash_attn_var_len` runtime (ms) under different sequence-length distributions with the same total tokens (64K).

| Input sequences | 1×64K | 4×16K | 8×8K | 16×4K | 32×2K | 64×1K |
|---|---|---|---|---|---|---|
| Runtime (ms) | 182.9 | 46.6 | 24.0 | 12.7 | 7.0 | 4.3 |

# C   ADDITIONAL COMMUNICATION OVERHEAD RESULTS

**SeqAlltoAll Micro-Benchmark.** We micro-benchmark FlashAttention2 GQA SeqAlltoAll communication time on two clusters with different bandwidth characteristics: H200 (900GB/s intra-node, 3200Gb/s inter-node) and H800 (400GB/s intra-node, 1600Gb/s inter-node). We use GQA with head_q=32, head_kv=8, and head_dim=128, and report the average AlltoAll latency (ms) under different sharding degrees (sp_size) and sequence lengths (seq_len). Table 4 shows a representative case with `seq_len`=262144 tokens, while Table 5 reports the full sweep.

Table 4: SeqAlltoAll micro-benchmark communication latency (ms) in FlashAttention2 GQA.

| **H200 Cluster** | | | **H800 Cluster** | | |
|---|---|---|---|---|---|
| **sp_size** | **seq_len** | **comm_avg_ms** | **sp_size** | **seq_len** | **comm_avg_ms** |
| 2 | 262144 | 23.401 | 2 | 262144 | 18.245 |
| 4 | 262144 | 16.107 | 4 | 262144 | 16.480 |
| 8 | 262144 | 8.513 | 8 | 262144 | 12.543 |
| 16 | 262144 | 14.639 | 16 | 262144 | 19.644 |
| 32 | 262144 | 11.030 | 32 | 262144 | 16.558 |

Table 5: Full SeqAlltoAll micro-benchmark communication time (ms) on H200 and H800 clusters.

| **H200 Cluster** | | | **H800 Cluster** | | |
|---|---|---|---|---|---|
| sp_size | seq_len | comm_avg_ms | sp_size | seq_len | comm_avg_ms |
| 2 | 131072 | 9.970 | 2 | 131072 | 11.079 |
| 2 | 262144 | 23.401 | 2 | 262144 | 18.245 |
| 2 | 524288 | 52.472 | 2 | 524288 | 60.709 |
| 4 | 131072 | 6.547 | 4 | 131072 | 6.943 |
| 4 | 262144 | 16.107 | 4 | 262144 | 16.480 |
| 4 | 524288 | 47.361 | 4 | 524288 | 44.372 |
| 8 | 131072 | 3.501 | 8 | 131072 | 6.360 |
| 8 | 262144 | 8.513 | 8 | 262144 | 12.543 |
| 8 | 524288 | 22.462 | 8 | 524288 | 28.686 |
| 8 | 1048576 | 64.878 | 8 | 1048576 | 119.293 |
| 16 | 131072 | 7.871 | 16 | 131072 | 12.333 |
| 16 | 262144 | 14.639 | 16 | 262144 | 19.644 |
| 16 | 524288 | 36.554 | 16 | 524288 | 41.430 |
| 16 | 1048576 | 68.558 | 16 | 1048576 | 104.011 |
| 32 | 131072 | 6.355 | 32 | 131072 | 17.219 |
| 32 | 262144 | 11.030 | 32 | 262144 | 16.558 |
| 32 | 524288 | 24.150 | 32 | 524288 | 31.089 |
| 32 | 1048576 | 48.356 | 32 | 1048576 | 69.296 |

