# OpenReview forum: "Scaling Large Vision-Language Model RL Training via Efficient Load Balancing"
_ICLR.cc/2026/Conference — ICLR 2026 Poster_

### Official Review · Reviewer_uQAx · 2025-10-16

**Soundness:** 3
**Presentation:** 2
**Contribution:** 2
**Rating:** 6
**Confidence:** 2

**Summary:**

This paper presents FlexRL, a system designed to address data loading bottlenecks and computational load imbalance in RL training for large multimodal models. Its core contributions are a decentralized data pipeline and a novel hybrid sequence sharding mechanism. The authors claim up to 4.2× speedup on a 128-GPU cluster.

**Strengths:**

- Accurately identifies key systemic bottlenecks in multimodal RL training.

- The proposed hybrid sequence sharding mechanism is ingenious and offers a promising direction for handling workload skew from heterogeneous sequences.

- The design of a dynamic execution engine to manage the scheduling complexity introduced by hybrid sharding is a significant engineering step.

**Weaknesses:**

- [Mandatory] There are minor typos, formatting inconsistencies, and grammatical errors. The authors should carefully proofread the manuscript.

- [Mandatory] The paper does not separately evaluate the individual contributions of the two core components: the "Decentralized Data Pipeline" and the "Hybrid Sequence Sharding." It is unclear which component drives the performance gains.

- [Mandatory] The hybrid sharding introduces complex All-to-All communication. While overlap strategies are qualitatively mentioned, a quantitative analysis of the communication overhead's impact on end-to-end performance under different cluster scales and network topologies is missing.

- [Mandatory] Key details regarding the scheduling heuristic and cost estimation model are missing, hindering reproducibility.

**Questions:**

Please refer to Weaknesses. Btw, I have some optional questions:

- [Optional] Dynamic sequence packing techniques, which concatenate short sequences into longer ones during training, have emerged recently. How does FlexRL fundamentally compare to such methods in terms of load balance efficiency and memory utilization? What are the relative advantages and disadvantages?

- [Optional] The title claims "Universal," but the method heavily relies on All-to-All communication within the attention mechanism. If future VLM architectures shift towards SSMs or other non-attention-based mechanisms, would FlexRL's core sharding mechanism remain effective? What is your view on this architectural dependency risk?

---

> ### Author Response · Authors · 2025-11-25
> **Rebuttal by Authors Part 1**
>
> Thank you for the thoughtful and constructive feedback. It has been highly valuable in improving our work. Below, we provide detailed responses to address each of your questions. We hope our responses have addressed your concerns, and we look forward to any further questions you may have!
>
> >### W1: Typos, formatting inconsistencies, and grammatical errors.
>
> We thank the reviewer for pointing this out. We will carefully proofread the manuscript and fix all typos, formatting inconsistencies, and grammatical issues.
>
> >### W2: Indivial Contributions of the two core components.
>
> Thank you for pointing this out. We agree that ablating the two main system components is important for understanding where the gains of our method come from. We have added a dedicated ablation study that separately evaluates the effects of **Decentralized Data Pipeline (DDP)** and **Hybrid Sequence Sharding (HSS)**.
>
> **Evaluation setup.** We use the same multimodal GRPO training workload as in W1, with **Qwen-2.5-7B-VL** and **FSDP = 32**. The baseline is the production **veRL + Bucket Balancing** system (with SP = 2). On top of this baseline, we evaluate: (i) **DDP only**, (ii) **HSS only**, and (iii) **FlexRL (DDP + HSS)**. The detailed runtimes and speedups for *update-actor*, *ref-log-prob*, *old-log-prob*, data loading, step time, and end-to-end throughput are as follows
>
> | Method | Update actor runtime (s) | Ref log prob runtime (s) | Old log prob runtime (s) | Data loading latency (s) | Step elapsed_time (s) | Overall Throughput (tokens/s) |
> | --- | --- | --- | --- | --- | --- | --- |
> | veRL + Bucket Balancing (Baseline) | 61.3 (1.00x) | 51.3 (1.00x) | 50.2 (1.00x) | 375.0 (1.00x) | 656.2 (1.00x) | 2975.0 (1.00x) |
> | + DDP only | 24.3 (2.52x) | 5.0 (10.26x) | 25.3 (1.99x) | 3.9 (96.15x) | 122.6 (5.35x) | 13905.0 (4.67x) |
> | + HSS only | 47.8 (1.28x) | 50.1 (1.02x) | 27.8 (1.81x) | 375.0 (1.00x) | 559.6 (1.17x) | 3489.0 (1.17x) |
> | **FlexRL (both enabled)** | **9.5 (6.45x)** | **3.7 (13.86x)** | **4.9 (10.24x)** | **3.2 (117.19x)** | **85.7 (7.66x)** | **22784.0 (7.66x)** |
>
> **Effect of DDP.** Enabling only **DDP** already yields large gains across the entire pipeline. Data-loading latency is reduced from 375.0 s to 3.9 s (96.15×), step time drops from 656.2 s to 122.6 s (5.35×), and overall throughput improves from 2.98k to 13.91k tokens/s (4.67×). This confirms that centralized multimodal I/O and preprocessing are a major bottleneck in RL training of VLMs, and that decentralizing these operations, plus removing redundant parameter/feature transmissions, substantially accelerates the system.
>
> **HSS-only analysis.** The relatively modest end-to-end speedup of **HSS-only** is due to where time is actually spent in our RL pipeline. The reported runtimes for *update-actor*, *ref-log-prob*, and *old-log-prob* include not only GPU computation, but also the **Ray-based transmission and launch overhead**. Without **DDP**, each RL task must transmit large vision-related data (e.g., image embeddings/features) from a central node before computation starts. This makes launching a single task cost on the order of hundreds of milliseconds. Moreover, Ray launches these tasks sequentially, so the effective startup time grows roughly linearly with the number of nodes and quickly becomes the dominant bottleneck. In this communication-bound regime, **HSS** does improve compute efficiency (e.g., *update-actor* and *old-log-prob* runtimes are reduced up to 1.8–2.0×), but these gains are largely hidden behind the much larger launch/communication overhead, resulting in only a 1.17× end-to-end speedup.
>
> **Combined effect (FlexRL).** In contrast, **DDP** brings benefits from two sources: (i) decentralized dataloading so that multimodal decoding and I/O happen locally, and (ii) elimination of repeated transmission of large vision features and related parameters across nodes. Once **DDP** removes this communication bottleneck, the workload becomes computation-bound, and **HSS** can fully take effect by balancing the sequence-level attention cost across GPUs.  Finally, when both components are enabled (**FlexRL**),  the system delivers 7.66× improvement over the baseline, demonstrating the efficiency of our method.

---

> ### Author Response · Authors · 2025-11-25
> **Rebuttal by Authors Part 2**
>
> >### W3-1:  Analysis of the communication overhead.
>
> - **SeqAlltoAll communication analysis.** In our setting, Ulysses-style sequence parallelism invokes a SeqAlltoAll primitive on the ulysses group. Let $d_{\text{sp}}$ denote the number of GPUs in this group. For each Q/K/V/output chunk $i$, the data size is $\text{Size}(i)$, and SeqAlltoAll performs an NCCL AlltoAll across $d_{\text{sp}}$ GPUs. The per-GPU AlltoAll traffic (forward + backward) is:
>
>     $$
>     \mathrm{AlltoAll\_Volume}
>     = \sum_{i \in \{q, k, v, \mathrm{out}\}}
>     \mathrm{Size}(i)\,
>     \frac{d_{\mathrm{sp}} - 1}{d_{\mathrm{sp}}}.
>     $$
>
>     As $d_{\text{sp}}$ increases, each GPU participates in more peers in the AlltoAll, so the term $(d_{\text{sp}}-1)/d_{\text{sp}}$ grows and the AlltoAll volume (and thus latency) increases.
>
> - **Communication overhead under different cluster scales and topologies**
>
>     Because the degree of head-parallel fashion sequence parallelism is limited by the number of attention heads (usually 32 or 64), we only scale to 4 nodes (32 GPUs). We perform analysis on two clusters:
>
>     |  | intra-node bandwidth | inter-node bandwidth |
>     | --- | --- | --- |
>     | H800 Cluster | 400GB/s | 1600Gb/s |
>     | H200 Cluster | 900GB/s | 3200Gb/s |
>     - **SeqAlltoAll communication**: We micro-benchmark SeqAlltoAll communication time with FlashAttention2 GQA (head_q=32, head_kv=8, head_dim=128) on different clusters. We report all2all communication times:
>
>     | **`H200 Cluster`** |  |  | **`H800 Cluster`** |  |  |
>     | --- | --- | --- | --- | --- | --- |
>     | sp_size | seq_len | comm_avg_ms | sp_size | seq_len | comm_avg_ms  |
>     | 2 | 131072 | 9.970 | 2 | 131072 | 11.079 |
>     | 2 | 262144 | 23.401 | 2 | 262144 | 18.245 |
>     | 2 | 524288 | 52.472 | 2 | 524288 | 60.709 |
>     | 4 | 131072 | 6.547 | 4 | 131072 | 6.943 |
>     | 4 | 262144 | 16.107 | 4 | 262144 | 16.480 |
>     | 4 | 524288 | 47.361 | 4 | 524288 | 44.372 |
>     | 8 | 131072 | 3.501 | 8 | 131072 | 6.360 |
>     | 8 | 262144 | 8.513 | 8 | 262144 | 12.543 |
>     | 8 | 524288 | 22.462 | 8 | 524288 | 28.686 |
>     | 8 | 1048576 | 64.878 | 8 | 1048576 | 119.293 |
>     | 16 | 131072 | 7.871 | 16 | 131072 | 12.333 |
>     | 16 | 262144 | 14.639 | 16 | 262144 | 19.644 |
>     | 16 | 524288 | 36.554 | 16 | 524288 | 41.430 |
>     | 16 | 1048576 | 68.558 | 16 | 1048576 | 104.011 |
>     | 32 | 131072 | 6.355 | 32 | 131072 | 17.219 |
>     | 32 | 262144 | 11.030 | 32 | 262144 | 16.558 |
>     | 32 | 524288 | 24.150 | 32 | 524288 | 31.089 |
>     | 32 | 1048576 | 48.356 | 32 | 1048576 | 69.296 |

---

> > ### Author Response · Authors · 2025-11-25
> >
> > >### W3-2: Analysis of the communication overhead.
> >
> > - **Communication overhead under different cluster scales and topologies**
> >
> >     Because the degree of head-parallel fashion sequence parallelism is limited by the number of attention heads (usually 32 or 64), we only scale to 4 nodes (32 GPUs). We perform analysis on two clusters:
> >
> >     |  | intra-node bandwidth | inter-node bandwidth |
> >     | --- | --- | --- |
> >     | H800 Cluster | 400GB/s | 1600Gb/s |
> >     | H200 Cluster | 900GB/s | 3200Gb/s |
> >     - **SeqAlltoAll communication**: We micro-benchmark SeqAlltoAll communication time with FlashAttention2 GQA (head_q=32, head_kv=8, head_dim=128) on different clusters.  We report all2all communication times:
> >
> >     | **`H200 Cluster`** |  |  | **`H800 Cluster`** |  |  |
> >     | --- | --- | --- | --- | --- | --- |
> >     | sp_size | seq_len | comm_avg_ms | sp_size | seq_len | comm_avg_ms  |
> >     | 2 | 131072 | 9.970 | 2 | 131072 | 11.079 |
> >     | 2 | 262144 | 23.401 | 2 | 262144 | 18.245 |
> >     | 2 | 524288 | 52.472 | 2 | 524288 | 60.709 |
> >     | 4 | 131072 | 6.547 | 4 | 131072 | 6.943 |
> >     | 4 | 262144 | 16.107 | 4 | 262144 | 16.480 |
> >     | 4 | 524288 | 47.361 | 4 | 524288 | 44.372 |
> >     | 8 | 131072 | 3.501 | 8 | 131072 | 6.360 |
> >     | 8 | 262144 | 8.513 | 8 | 262144 | 12.543 |
> >     | 8 | 524288 | 22.462 | 8 | 524288 | 28.686 |
> >     | 8 | 1048576 | 64.878 | 8 | 1048576 | 119.293 |
> >     | 16 | 131072 | 7.871 | 16 | 131072 | 12.333 |
> >     | 16 | 262144 | 14.639 | 16 | 262144 | 19.644 |
> >     | 16 | 524288 | 36.554 | 16 | 524288 | 41.430 |
> >     | 16 | 1048576 | 68.558 | 16 | 1048576 | 104.011 |
> >     | 32 | 131072 | 6.355 | 32 | 131072 | 17.219 |
> >     | 32 | 262144 | 11.030 | 32 | 262144 | 16.558 |
> >     | 32 | 524288 | 24.150 | 32 | 524288 | 31.089 |
> >     | 32 | 1048576 | 48.356 | 32 | 1048576 | 69.296 |
> >     - **End-to-end comparison**: We know that using higher SP also delivers better load balancing, but with higher communication overhead. So we further evaluate the efficiency of FlexRL on addressing this problem.
> >         - **Testbed:** We test on 8 H800.
> >         - **Model:** Qwen2.5-7B-VL
> >         - **Dataset:** Geo3K + NExTQA + LongVILA-Reason.
> >             - Sampling weight: [5:2:1]
> >             - `global_batch_size`: 32
> >             - `max_frame_number` : 128
> >             - `max_response_length` : 256.
> >         - **Parallelism**: TP=2 for rollout and FSDP=8 for model forward and update_policy. Both with the Decentralized Data Pipeline enabled to remove the impact of data transfer.
> >             - FlexRL(w/ vanilla SP) : sharding degree {2, 4, 8}
> >             - FlexRL(w/ DSS):  sharding degree range from 1~8
> >
> >         | `On H200 Cluster` | Update actor runtime (s) | Ref log prob runtime (s) | Old log prob runtime (s) | Overall Throughput (tokens/s/GPU) | Balance Ratio (→1.0 better) |
> >         | --- | --- | --- | --- | --- | --- |
> >         | verl (baseline) | 5.90 (1.00x) | 1.89 (1.00x) | 1.89 (1.00x) | 155 (1.00x) | 6.7 |
> >         | FlexRL (SP=2) | 4.43 (1.33x) | 1.88 (1.01x) | 2.49 (0.76x) | 162 (1.05x) | 3.82  |
> >         | FlexRL (SP=4) | 4.33 (1.36x) | 1.94 (0.97x) | 3.19 (0.59x) | 173 (1.12x) | 1.12  |
> >         | FlexRL (SP=8) | 5.81 (1.02x) | 1.25 (1.51x) | 1.97 (0.96x) | 175 (1.13x) | 1.00 |
> >         | FlexRL (w/ DSS) | **3.52 (1.68x)** | **1.10 (1.72x)** | **1.70 (1.11x)** | **232 (1.50x)** | **1.00** |
> >
> >         | `On H800 Cluster` | Update actor runtime (s) | Ref log prob runtime (s) | Old log prob runtime (s) | Overall Throughput (tokens/s/GPU) | Balance Ratio (→1.0 better) |
> >         | --- | --- | --- | --- | --- | --- |
> >         | verl (baseline) | 6.24 (1.00x) | 1.59 (1.00x) | 1.60 (1.00x) | 103 (1.00x) | 6.7 |
> >         | FlexRL (SP=2) | 5.13 (1.22x) | 1.17 (1.36x) | 1.16 (1.38x) | 123 (1.19x) | 3.82 |
> >         | FlexRL (SP=4) | 4.41 (1.41x) | 1.05 (1.51x) | 1.80 (0.89x) | 153 (1.49x) | 1.12 |
> >         | FlexRL (SP=8) | 5.81 (1.07x) | 1.45 (1.10x) | 2.46 (0.65x) | 141 (1.37x) | 1.00 |
> >         | FlexRL (w/ DSS) | **3.90 (1.60x)** | **0.90 (1.77x)** | **2.01 (0.80x)** | **172 (1.67x)** | **1.00** |
> >
> > > Balance ratio = max(gpu_attn_flops) / mean(sum(gpu_attn_flops))
> >
> > We see that simply increasing the Ulysses-SP degree trades better load balance for growing communication overhead. SP=2 and SP=4 steadily improve both per-stage runtimes and throughput while improve the balance ratio. When SP=8, workload is balanced, but the efficiency start drops significantly, indicating that extra All-to-All communication has begun to dominate. Reducing communication and fine-grained overlapping are effective ways.
> >
> > In contrast, **FlexRL with DSS** achieves both near-perfect balance (ratio 1.0) and strictly better end-to-end efficiency. The e2e throughput reaches **172 tokens/s/GPU (1.67× improvement over baseline).** This shows that our method can reap the benefits of sequence sharding and communication overlapping for concurrently balancing workload and reducing communication overhead.

---

> ### Author Response · Authors · 2025-11-25
> **Rebuttal by Authors Part 4**
>
> >### W4-1: Detailed scheduling heuristic and cost estimation model
>
> Thank you for pointing out that the description of our scheduler and cost model was too high-level. We agree that this is important for reproducibility and will make the implementation details explicit in the revised version.
>
> ### Problem formulation
>
> We consider one training step over a global mini-batch of $N$ sequences. Each sequence $i \in \{1,\dots,N\}$ has length $h_i$, which already includes both textual and visual tokens. We want to schedule this batch over $D$ devices (ranks) using hybrid sequence sharding.
>
> ### Decision variables
>
> - For each sequence $i$, we choose a sharding degree, meaning that sequence $i$ is distributed into $p_i$ equal shards, each carrying $h_i / p_i$ tokens.
>
>     $$
>     p_i \in \{1,2,4,\dots,p_{\max}\}, \qquad p_{\max} \le D,
>     $$
>
> - We assign these shards to devices. Let $y_{i,d} \in \{0,1\}$ indicate whether device $d$ holds one shard of sequence $i$. We require
>
>     $$
>     \sum_{d=1}^D y_{i,d} = p_i \quad \forall i,
>     $$
>
>      so that each sequence is sharded across exactly $p_i$ distinct devices.
>
>
> ### Per-device load
>
> On device $d$, we track two quantities:
>
> - **Memory load**
>
>     $$
>     M_d = \sum_{i=1}^N \frac{h_i}{p_i} \, y_{i,d},
>     $$
>
>     i.e., the total number of tokens stored on device $d$.
>
> - **Computation load**
>
>     We focus on the attention computation. For full self-attention the total FLOPs scale as $O(h_i^2)$. Under sequence sharding with degree $p_i$, each device processes roughly a $1/p_i$ fraction of this work, so the per-device contribution from sequence $i$ is proportional to $h_i^2 / p_i$. The total compute load on device $d$ is therefore
>
>     $$
>     C_d = \sum_{i=1}^N \frac{h_i^2}{p_i} \, y_{i,d}.
>     $$
>
>     We denote the ideal per-device averages (for a given layout) as
>
>     $$
>     M^\star = \frac{1}{D}\sum_{i=1}^N h_i,\qquad C^\star = \frac{1}{D}\sum_{i=1}^N {h_i^2},
>     $$
>
>     and set a hard memory threshold
>
>     $$
>     M_{\text{th}} = (1+\epsilon)\, M^\star,\quad \epsilon=0.1.
>     $$
>
>
> ### Communication cost (sharding overhead)
>
> We model the communication overhead of Ulysses-style sequence parallelism following the SeqAlltoAll analysis in LoongTrain. For a sequence of length $h$ sharded across $p$ devices, the communication volume per device scales as
>
> $$
> \text{vol}(h,p) \propto h \cdot \frac{p-1}{p^2}.
> $$
>
> We incorporate the cluster topology by distinguishing intra-node versus inter-node traffic. Let $G=8$ be the number of GPUs per node. For a sharding degree $p$, we define the fraction of communication that can be kept intra-node as
>
> $$
> f_{\text{intra}}(p) =
> \begin{cases}
> 1, & p \le G,
> \dfrac{G-1}{p-1}, & p > G,
> \end{cases}
> \qquad
> f_{\text{inter}}(p) = 1 - f_{\text{intra}}(p).
> $$
>
> The per-sequence communication cost is then
>
> $$
> \text{Comm}(h,p) =
> h \cdot \frac{p-1}{p^2}
> \left(
> \frac{f_{\text{intra}}(p)}{B_{\text{intra}}}+\frac{f_{\text{inter}}(p)}{B_{\text{inter}}}
> \right),
> $$
>
> where $B_{\text{intra}}$ and $B_{\text{inter}}$ are the effective intra- and inter-node bandwidths, respectively. In our cluster we normalize $B_{\text{inter}} = 1$ and $B_{\text{intra}} = 16$.
>
> The total sharding cost of a layout is
>
> $$
> C_{\text{shard}} = \sum_{i=1}^N \text{Comm}(h_i, p_i).
> $$
>
> This cost model is used to rank candidate layouts. The actual runtime additionally includes the compute time, which we measure separately.
>
> ### Objective and constraints
>
> The scheduling problem is a discrete, multi-objective load-balancing problem.
>
> - **Hard constraint (memory):**
>
> $$
> M_d \le M_{\text{th}} \quad \forall d.
> $$
>
> - **Soft objectives:**
> Among all layouts satisfying the memory constraint, we want to
>     1. balance the computation across devices, and
>     2. minimize the total sharding overhead.
>
> Formally, we use an objective:
>
> $$
> \min_{\{p_i, y_{i,d}\}}\max_{d} C_d,
> $$
>
> i.e., we primarily minimize the maximum compute load across devices.

---

> ### Author Response · Authors · 2025-11-25
> **Rebuttal by Authors Part 5**
>
> >### W4-1: Detailed scheduling heuristic and cost estimation model
>
> ### Scheduling heuristic
>
> Our greedy heuristic starts from an over-sharded, nearly balanced layout and progressively merges shards to reduce $C_{\text{shard}}$ while maintaining the memory and compute balance constraints.
>
> - **Input**:
>     - Sequence lengths $\{h_i\}_{i=1}^N$;
>     - Number of devices $D$, per-device token budget $M_{\text{th}}$;
>     - Effective bandwidths $B_{\text{intra}}, B_{\text{inter}}$;
>     - Candidate sharding degrees $p_i \in \{1,2,4,\dots,p_{\max}\}$ (powers of two).
> - **Output**: Per-sequence sharding degree $p_i$ and shard placement $\{y_{i,d}\}$.
> - **Objective**: Minimize the maximum runtime across all devices, using the communication model as a proxy for the sharding overhead.
>
> We use three stages:
>
> 1. **Initialization. (load balancing with fine grained sharding)**
>
>     We sort sequences by length in descending order. For each sequence $i$, we compute an initial sharding degree by dividing a roughly fixed shard length $k$. We then greedily assign the $p_i$ shards of sequence $i$ to the subset of devices with sufficient remaining memory budget (respecting $M_d \le M_{\text{th}}$) and the lowest current compute load $C_d$.
>
> 2. **Shard refinement. (Reducing all2all overhead)**
>
>     We iteratively reduce sharding degrees from $p_{\max}$ down to $2$. At each level $p$, we consider sequences with current degree $p_i = p$. For two sequences $i$ and $j$ that are sharded over the same device group and have similar lengths, we simulate *merging* their sharding degree from $p$ to $p/2$ (i.e., reducing the number of shards while keeping the device group fixed). Among all valid merges that respect the memory constraint and do not introduce excessive compute imbalance, we greedily commit the merge that yields the largest reduction in $C_{\text{shard}}$. We repeat this process until no valid candidate merges remain at this level, then move to the next lower sharding degree.
>
> 3. **Short sequence reordering. (Further reducing all2all overhead)**
> Finally, we collect sequences with $p_i \le 2$ and $h_i < M^\star / 4$. For these short sequences, we try to set $p_i = 1$ and re-assign them using a greedy “first-fit-decreasing” style bucketing: we process sequences in descending $h_i$, and place each onto the device that (i) remains under the memory threshold and (ii) yields the smallest increase in compute load $C_d$. This step further reduces communication overhead by avoiding unnecessary sharding of short sequences.
>
> We will provide pseudocode, the concrete hyperparameters (memory threshold, imbalance tolerance $\epsilon = 0.1$, shard length constant $k$), and an open-source implementation in the appendix for reproducibility.

---

> ### Author Response · Authors · 2025-11-25
> **Rebuttal by Authors Part 6**
>
> >### Q1: Compare with Dynamic sequence packing.
>
> We thank the reviewer for raising the question.
>
> - **Load-balance efficiency.**
>
>     Dynamic sequence packing typically combines (i) greedy length-based bucketing and (ii) reordering/concatenating short sequences into longer packed sequences so that each DP rank runs a global masked attention over its local packed tokens. This works well when sequence lengths are not extremely skewed, as stated in Sec-3 of the paper. This works in pretraining, but is not the case in the VLM RL regime, where sequence lengths are *extremely* heterogeneous (short text, long reasoning, and very long video contexts in the same batch).
>
>     ```
>     Batch A:
>     	DP0: [    32K    ]**(Straggler)**
>     	DP1: [8K|8K|8K|8K]
>     	DP2: [8K|8K|8K|8K]
>     	DP3: [8K|8K|8K|8K]
>
>     Batch B:
>     	DP0: [    32K    ]**(Straggler)**
>     	DP1: [    32K    ]
>     	DP2: [    32K    ]
>     	DP3: [    32K    ]
>     ```
>     A: Even with dynamic packing, rank 0 inevitably holds the 32K sequence. Other ranks are packed to 32K, so the step still behaves like the worst-case DP instance (Batch B). DP 0 becomes the ***straggler***.
>
>     FlexRL’s Hybrid Sequence Sharding (HSS) directly targets this failure mode by allowing each sequence to use its own sharding degree and device group, and by sharding long sequences at sub-sequence granularity. A 32K outlier can be split across multiple GPUs, while shorter sequences stay unsharded, so both compute and memory per GPU are balanced without being bound by a single long sequence.
>
> - **Memory utilization.**
>
>     In terms of memory usage, both DSP and FlexRL aim to pack tokens up to the per-GPU capacity, so the peak memory *budget* is comparable. The key difference is *where* the long-sequence activations live:
>
>     - In DSP, the entire long sequence must reside on a single GPU for attention, which (i) can trigger OOM when the context length exceeds that GPU’s capacity, and (ii) makes that GPU the memory and compute bottleneck.
>     - In FlexRL, HSS shards long sequences across multiple GPUs, so no single device needs to store all activations of an extreme outlier; this avoids the OOM behavior of sequence-level methods and leads to more uniform memory pressure across GPUs.
>
>     Empirically, we observe similar average memory utilization, but substantially fewer OOMs and less skew in peak memory across ranks when using HSS instead of pure packing.
>
> - **Empirical comparison.**
>     To quantify this, we run a controlled micro-benchmark training Qwen-2.5-VL-7B on 16 A800 GPUs. We compare FSDP + sequence parallelism (SP) with and without dynamic sequence packing, and with our HSS, under three workload patterns:
>
>     - **C1 (balanced short sequences):** All ranks get 32 / 64 / 128 sequences of length 1K. This is a very light workload because sequences are all short. Simply packing can achieve load balancing.
>     - **C2 (all long sequences):** All ranks get one long sequence of length 32K / 64K / 128K. This is the worst case
>     - **C3 (imbalanced mix):** Rank 0 gets one long 32K / 64K / 128K sequence, the other ranks get 32 / 64 / 128 short 1K sequences.
>
>     |  | 32K (SP=1) | 64K (SP=2) | 128K (SP=4) |
>     | --- | --- | --- | --- |
>     | *C1*  | *3203* | *2797* | *2801* |
>     | C2 | 1609  | 1405  | 886  |
>     | **C3 (w/ Packing)** | **1615** (1.00x) | **1402** (1.00x) | **891** (1.00x) |
>     | **C3 (w/ ours)** | **2998 (1.81x)** | **2703 (1.92x)** | **2748 (3.10x)** |
>   The results in the table show that:
>
> - **Upper bound (C1).** When all ranks see many short sequences, and there is no severe length skew, we get the best throughput (C1).
> - **Skewed workload issue (C2):** C2 shows the worst case where all sequences are long sequences. The throughput drops by 50%~69%
> - **Packing (C3 + Packing).**   C1+only one sequence, with sequence packing, yields almost no improvement compared to even C2, which is the worst case. The single long sequence (**Straggler**) still dominates the step time.
> - **Ours (C3 + ours).** With our dynamic sharding, the workload in the imbalanced C3 setting recovers most of the gap to the balanced upper bound C1. Our method delivers 1.81~3.10x speed up compared to packing.
>
> - Disadvantages and advantages
>     - **Dynamic sequence packing (DSP)**
>         - **Advantages:**
>             - Simple to add on top of existing DP/SP.
>             - Effectively reduces padding
>         - **Disadvantages:**
>             - Operates at sequence granularity: cannot remove stragglers when the length is highly skewed
>             - Long sequences must still fit on one GPU, causing memory hotspots and potential OOM.
>     - **FlexRL with Hybrid Sequence Sharding (HSS)**
>         - Operates at sub-sequence granularity, eliminating stragglers.
>         - Distributes long-sequence activations across GPUs, balancing memory and reducing OOM.

---

> ### Author Response · Authors · 2025-11-25
> **Rebuttal by Authors Part 7**
>
> ## Q2: Applicability to SSM and Other Non-Attention Architectures
>
> Thank you for raising this point. We’re happy to clarify. Our method does not rely on softmax attention itself, but only on the ability to partition computation along the head dimension. As long as the underlying architecture exposes multiple independent heads (or channels that play the same role). FlexRL’s hybrid sequence sharding and head-parallel scheduling remain directly applicable.
>
> In current practice, even non-standard or hybrid architectures retain explicit multi-head structure, including SSM and linear-attention variants. For example:
>
> - Qwen3-Next-80B-A3B-Instruct [1] (Linear Attention + Attention): 16 attention heads, and its linear-attention component is also multi-head, where linear_`key_heads = 16`, linear_`value_heads = 32`.
> - Kimi-Linear-48B-A3B-Instruct [2] (Linear Attention + Attention): 32 heads in the linear/KDA block and 32 standard attention heads.
> - Nemotron-H-8B-Base-8K [3] (Mamba + attention): 128 Mamba heads and 32 standard attention heads.
>
> These cases show that attention variants are still organized in a multi-head fashion. Under such designs, FlexRL can continue to shard and schedule per-head computations exactly as in the standard attention case.
>
> Thus, our method can naturally generalize to other attention variants. The only change required is the comp/mem model for the specific operator into our heuristic algorithm. The hybrid sequence sharding strategy and execution engine remain unchanged.
>
> [1] [Qwen3-Next-80B-A3B-Instruct](https://huggingface.co/collections/Qwen/qwen3-next)
>
> [2] [Kimi-Linear-48B-A3B-Instruct ](https://huggingface.co/moonshotai/Kimi-Linear-48B-A3B-Instruct)
>
> [3] [Nemotron-H-8B-Base-8K](https://huggingface.co/nvidia/Nemotron-H-8B-Base-8K)

---

### Official Review · Reviewer_cr9u · 2025-10-31

**Soundness:** 3
**Presentation:** 3
**Contribution:** 3
**Rating:** 6
**Confidence:** 4

**Summary:**

This paper presents a distributed training pipeline for RL training stage of large-scale VLMs involving multimodal data (images, videos, texts). The key challenge in the multimodal RL training is the highly diverse data length (short text, long text, long image&video tokens), which is hard to be properly scheduled in a distributed training system. It mainly proposes a decentralized data pipeline to properly schedule the data with a single controller, and a hybrid sequence sharding technique to partition sequences into finegrained chunks to enable sub-sequence level load balancing. Existing technique Ulysses Sequence Parallelism is used to enable sequence parallel training.

**Strengths:**

-	The proposed hybrid sharding technique is novel and alleviate the issue of imbalanced loading.
-	Experiments show the proposed approach outperform the speed of existing approach such as verl on video understanding tasks.

**Weaknesses:**

-	The speed of running a batch on a gpu should be clarified more. How does the gpu handles sequence with different length in a batch? From figure2, a gpu will pack samples with different lengths into groups and conduct their attention operation separately. While in some implementations using generic sequence packing and masked attention, the running time is irrelevant to the sequence length of each sample since a global masked attention of all tokens in a batch is conducted. How much speed up does the proposed approach have compared to the global-attention method?
-	The paper only provides numbers for speed, while accuracy numbers are not provided. The accuracies of using the proposed approach is also required to show the method’s robustness for different applications.

**Questions:**

see above

---

> ### Author Response · Authors · 2025-11-25
> **Rebuttal by Authors Part 1**
>
> Thank you for the thoughtful and constructive feedback. It has been highly valuable in improving our work. Below, we provide detailed responses to address each of your concerns.
>
> >### W1: How much speed up does the proposed approach have compared to the global-attention method?
>
> Great question. Actually, the runtime when using sequence packing is related to the length distribution of packed sequences:
>
> - `flash_attn_var_len` sequence packing analysis
>
>     FlashAttention uses `flash_attn_var_len` to implement https://github.com/Dao-AILab/flash-attention/issues/654, which is mathematically equal to usinga  global attention mask. The computation is actually less than full attention with computation complexity $O(N^2)$. The computation is $k\sum^{i=0}_{i<s}{l_i^2}$, wher $l_i$ is the length of sequence $i$, k is a multiplier. The length distribution of all sequences would significantly affet the runntime.
>
> We do an experiment on H800 GPUs to demonstrate it. We use  `flash_attn_var_len` to perform GQA (qhead=64, kvhead=8, head_dim=128) and measure the runtime of different inputs distribution. The input is a batch of sequences with same length. The sum of tokens is 128K and the only difference is the number of tokens and sequence length. The result is as follows:
>
> | Input Seq | 1 $\times$  64K | 4 $\times$  16K | 8 $\times$  8K | 16 $\times$  4K | 32 $\times$  2K | 64 $\times$  1K |
> | --- | --- | --- | --- | --- | --- | --- |
> | FA Runtime(ms) | 182.9 | 46.6 | 24.0 | 12.7 | 7.0 | 4.3 |
> - **Attention computation speedup:** We run a benchmark on training Qwen-2.5-VL 7B with 16 A800 GPUs.
>
>     We set the input data to the following settings
>
>     - **C1 (balanced short sequences):** All ranks get 32 / 64 / 128 sequences of length 1K.
>     - **C2 (all long sequences):** All ranks get one long sequence of length 32K / 64K / 128K.
>     - **C3 (imbalanced mix):** Rank 0 gets one long 32K / 64K / 128K sequence, the other ranks get 32 / 64 / 128 short 1K sequences.
>
>     For each configuration, we run 20 forward+backward steps and report the average throughput in tokens / GPU / second. We consider three SP degrees (SP=1,2,4) corresponding to max context lengths 32K, 64K, 128K:
>
>     |  | 32K (SP=1) | 64K (SP=2) | 128K (SP=4) |
>     | --- | --- | --- | --- |
>     | *C1* | *3203* | *2797* | *2801* |
>     | C2 | 1609  | 1405  | 886  |
>     | **C3 (w/ Packing)** | **1615** (1.00x) | **1402** (1.00x) | **891** (1.00x) |
>     | **C3 (w/ ours)** | **2998 (1.81x)** | **2703 (1.92x)** | **2748 (3.10x)** |
>
>     The result shows that our method efficiently addresses the imbalance issue of vanilla packing.
>
>     - **Upper bound (C1).** When all ranks see many short sequences, and there is no severe length skew, we get the best throughput (C1).
>     - **Skewed workload issue (C2):** C2 shows the worst case where all sequences are long sequences. The throughput drops by 50%~69%
>     - **Packing (C3 + Packing).**   C1+only one sequence, with sequence packing, yields almost no improvement compared to even C2, which is the worst case. The single long sequence (**Straggler**) still dominates the step time.
>     - **Ours (C3 + ours).** With our dynamic sharding, the workload in the imbalanced C3 setting recovers most of the gap to the balanced upper bound C1. Our method delivers 1.81~3.10x speed up compared to packing.
> - **E2E Speedup Compared to global attention:**  We compare FlexRL against the veRL baseline under the same RL training (GRPO) configuration:
>     - **Testbed:** 32 H800 GPUs
>     - **Model:** Qwen-2.5-7B-VL
>     - **Dataset:** LongVILA-Reason.
>         - `global_batch_size`: 256. Length of each sample is about 8K, domen
>         - `max_frame_number` We constantly sample 128 frames for each data item.
>         - `max_response_length` : 256.
>     - **Parallelism**: TP=4 for rollout and FSDP=32 for model forward and update_policy.
>         - Baseline: SP=1
>         - FlexRL: HSS shard degree: 1~4
>
>     | Method | Update actor runtime (s) | Ref log prob runtime (s) | Old log prob runtime (s) |
>     | --- | --- | --- | --- |
>     | veRL + Packing (Masked attention) | 13.2 (1.00x) | 9.07 (1.00x) | 9.09 (1.00x) |
>     | **FlexRL w/ speedup** | **5.39 (2.48x)** | **3.79 (2.39x)** | **3.63 (2.50x)** |
>
>     This end-to-end comparison shows that, under the same RL configuration, FlexRL accelerates all three expensive RL stages by roughly a factor of 2.4–2.5× over the veRL baseline with global-attention-style packing.

---

> ### Author Response · Authors · 2025-11-25
> **Rebuttal by Authors Part 2**
>
> >### Q2: The paper only provides numbers for speed, while accuracy numbers are not provided.
>
> Thank you for pointing out the lack of accuracy verification. To fully address your concern, we have run an RL training recipe (Qwen2.5-7B-VL on GRPO+NExTQA) for 150 steps under both the veRL baseline and FlexRL, and we provide a [figure link](https://anonymous.4open.science/r/rebuttal-8182/nextqa_mean_reward_score.pdf) to the training curves of `mean_critic_score`. These curves show that our approach improves training efficiency while maintaining essentially the same convergence behaviour as the baseline. We will add this figure and accuracy verification to the main paper.
>
> We hope our responses have addressed your concerns, and we look forward to any further questions you may have!

---

> > ### Comment · Reviewer_cr9u · 2025-11-25
> >
> > Thanks for the rebuttal. My concerns have been addressed. I will raise my score.

---

> > > ### Author Response · Authors · 2025-11-26
> > >
> > > We sincerely thank you for your kind response.  We deeply appreciate your recognition of our work and your decision to update the review score, which truly means a lot to us.

---

### Official Review · Reviewer_apGj · 2025-11-03

**Soundness:** 2
**Presentation:** 2
**Contribution:** 2
**Rating:** 4
**Confidence:** 2

**Summary:**

FlexRL is an end-to-end optimization system built on the verl framework to improve the efficiency of RL training for MLLMs. It addresses two primary bottlenecks: (1) a Decentralized Data Pipeline that distributes multimodal data loading and preprocessing across worker nodes while the control node handles only lightweight metadata, eliminating centralized I/O bottlenecks; and (2) a Hybrid Sequence Sharding mechanism that partitions sequences into fine-grained chunks to achieve subsequence-level load balancing, mitigating uneven GPU utilization caused by extreme length disparities across modalities such as text, images, and video.

**Strengths:**

The paper systematically analyzes practical bottlenecks across the entire RL training pipeline for MLLMs rather than focusing on a single stage, and it demonstrates strong system-level completeness.

**Weaknesses:**

I must first note that I am not very familiar with mlsys, while I only offer a limited perspective on this paper.

1. Why not compare against other verl-based optimized frameworks, such as [1], which also targets long-video scenarios?
2. The paper lacks concrete ablations; for example, it does not separately quantify the contributions of the Decentralized Data Pipeline and the Hybrid Sequence Sharding components.
3. Is the framework primarily intended for highly imbalanced workloads? The design appears to degenerate to conventional parallelism, but comparisons under balanced workloads (e.g., image-only or pure-text) are missing.
4. Performance under different settings is not reported, e.g., varying batch size and tensor/pipeline/sequence parallelism (TP/PP/SP) sizes.
5. As an mlsys work, the paper does not provide an engineering code release or community usage feedback, which I consider a weakness.

[1] Scaling RL to Long Videos. NeurIPS 2025.

**Questions:**

N/A

---

> ### Author Response · Authors · 2025-11-25
> **Rebuttal by Authors Part 1**
>
> Thank you for your valuable and constructive feedback, which contributes a lot to the enhancement of our paper. Below, we provide our responses addressing your concerns. And all the revisions will be updated in the forthcoming rebuttal revision version of the paper.
>
> >### Q1: Why not compare against other verl-based optimized frameworks[1]?
>
>
> Thank you for pointing out **LongRL**[1], which is indeed a VeRL-based optimized framework for long-video RL. LongRL mainly targets long video training, making efforts on long sequence training rather than addressing load imbalance. The authors of LongRL have also explicitly acknowledged [2] that their system does not currently optimize for mixed datasets with extreme sequence length variations. This limitation further underscores the importance of FlexRL, which is specifically designed to handle such highly imbalanced workloads efficiently. We have added this  baseline and compared it directly with FlexRL:
>
> We compare FlexRL against LongRL under the same RL training (GRPO) configuration:
>
> - **Testbed:** We test on 128GPU (16 $\times$ 8 H800).
> - **Model:** Qwen-2.5-7B-VL
> - **Dataset:** Geo3K + NExTQA + LongVILA-Reason.
>     - Sampling weight: [5:2:1]
>     - `global_batch_size`: 512, total ~2M tokens
>     - `max_frame_number` : 256
>     - `max_response_length` : 256. We set a small value. Rollout optimization is another big topic.
> - **Parallelism**: TP=4 for rollout and FSDP=32 for model forward and update_policy.
>     - FlexRL: HybridSP (shard degree: 1~4)
>     - LongRL: MMSP(size=8)
>
> |  | Update actor runtime (s) | Ref log prob runtime (s) | Old log prob runtime (s) | Data loading latency (s) | Overall Throughput (tokens/s) |
> | --- | --- | --- | --- | --- | --- |
> | LongRL | 18.1 (1.00x) | 4.6 (1.00x) | 23.7 (1.00x) | 356.7 (1.00x) | 4282 |
> | Ours | **9.5 (1.91x)** | **3.7 (1.24x)** | **4.9 (4.84x)** | **3.2 (117.19x)** | **22784 (5.32x)** |
>
> The results show that FlexRL achieves significant speedups across all stages. Compared to LongRL, FlexRL delivers a **5.32× overall speedup**. Specifically, FlexRL **eliminates the I/O bottleneck** through the Decentralized Data Pipeline (DDP), reducing data loading latency from 356.7 s to 3.2 s. In compute-intensive stages such as *Update actor* and *Old log prob*, FlexRL achieves **1.91×–4.84×** speedups. These results demonstrate that FlexRL’s Hybrid Sequence Sharding (HSS) handles load imbalance from extreme sequence length variations (image–text mixtures) much more effectively than LongRL’s MM-SP strategy.
>
> **[1] Scaling RL to Long Videos. NeurIPS 2025.**
>
> **[2] [Sequence length unbalance. LongRL GitHub Issue #3](https://github.com/NVlabs/Long-RL/issues/3).**

---

> ### Author Response · Authors · 2025-11-25
> **Rebuttal by Authors Part 2**
>
> >### Q2: Comprehensive E2E evaluation and Ablations Study
>
> Thank you for pointing this out. We agree that ablating the two main system components is important for understanding where the gains of our method come from. We have added a dedicated ablation study that separately evaluates the effects of **Decentralized Data Pipeline (DDP)** and **Hybrid Sequence Sharding (HSS)**.
>
> **Evaluation setup.** We use the same multimodal GRPO training workload as in W1 on 128 H800s, with **Qwen-2.5-7B-VL** and **FSDP = 32**. The baseline is the production **veRL + Bucket Balancing** system (with SP = 2). On top of this baseline, we evaluate: (i) **DDP only**, (ii) **HSS only**, and (iii) **FlexRL (DDP + HSS)**. The detailed runtimes and speedups for *update-actor*, *ref-log-prob*, *old-log-prob*, data loading, step time, and end-to-end throughput are as follows
>
> | Method | Update actor runtime (s) | Ref log prob runtime (s) | Old log prob runtime (s) | Data loading latency (s) | Step elapsed_time (s) | Overall Throughput (tokens/s) |
> | --- | --- | --- | --- | --- | --- | --- |
> | veRL + Bucket Balancing (Baseline) | 61.3 (1.00x) | 51.3 (1.00x) | 50.2 (1.00x) | 375.0 (1.00x) | 656.2 (1.00x) | 2975.0 (1.00x) |
> | + DDP only | 24.3 (2.52x) | 5.0 (10.26x) | 25.3 (1.99x) | 3.9 (96.15x) | 122.6 (5.35x) | 13905.0 (4.67x) |
> | + HSS only | 47.8 (1.28x) | 50.1 (1.02x) | 27.8 (1.81x) | 375.0 (1.00x) | 559.6 (1.17x) | 3489.0 (1.17x) |
> | **FlexRL (both enabled)** | **9.5 (6.45x)** | **3.7 (13.86x)** | **4.9 (10.24x)** | **3.2 (117.19x)** | **85.7 (7.66x)** | **22784.0 (7.66x)** |
>
> **Effect of DDP.** Enabling only DDP already yields large gains across the entire pipeline. Data-loading latency is reduced from 375.0 s to 3.9 s (96.15×), step time drops from 656.2 s to 122.6 s (5.35×), and overall throughput improves from 2.98k to 13.91k tokens/s (4.67×). This confirms that centralized multimodal I/O and preprocessing are a major bottleneck in RL training of VLMs.
>
> **HSS-only analysis.** The relatively modest end-to-end speedup of HSS-only is due to where time is actually spent in our RL pipeline. The reported runtimes for *update-actor*, *ref-log-prob*, and *old-log-prob* include not only GPU computation, but also the Ray-based transmission and launch overhead. Without DDP, each RL task must transmit large vision-related data (e.g., image embeddings/features) from a central node before computation starts. This makes launching a single task cost on the order of hundreds of milliseconds. Moreover, Ray launches these tasks sequentially, so the effective startup time grows roughly linearly with the number of nodes and quickly becomes the dominant bottleneck. In this communication-bound regime, HSS does improve compute efficiency (e.g., *update-actor* and *old-log-prob* runtimes are reduced up to 1.8–2.0×), but these gains are largely hidden behind the much larger launch/communication overhead, resulting in only a 1.17× end-to-end speedup.
>
> **Combined effect (FlexRL).** In contrast, DDP brings benefits from two sources: (i) decentralized dataloading so that multimodal decoding and I/O not happen only on the master node, and (ii) elimination of repeated transmission of large vision features and related parameters across nodes. Once DDP removes this communication bottleneck, the workload becomes computation-bound, and HSS can fully take effect by balancing the sequence-level attention cost across GPUs. When both components are enabled (**FlexRL**), and throughput increases by 7.66×.
>
> This response also presents as a comprehensive E2E evaluation. Our new results show that our method delivers up to 6.45x speedup on update actor runtime and **7.66x improvement** on e2e  throughput. We will add these results to the paper.

---

> ### Author Response · Authors · 2025-11-25
> **Rebuttal by Authors Part 3**
>
> >### Q3: Comparisons under balanced workloads.
>
> You are right! When handling a balanced workload, our method **degenerates to conventional parallelism (sequence packing with greedy bucketing).** We compare FlexRL against the veRL baseline under the following RL training (GRPO) configuration and report the speedup across all stages of RL training:
>
> - **Testbed:** 32 H800 GPUs
> - **Model:** Qwen-2.5-7B-VL
> - **Dataset:** LongVILA-Reason.
>     - `global_batch_size`: 256. all sequences ≈8K tokens
>     - `max_frame_number` We constantly sample 128 frames for each video.
>     - `max_response_length` : 256.
> - **Parallelism**: TP=4 for rollout and FSDP=32 for model forward and update_policy.
>     - Baseline: SP=1
>     - FlexRL: HSS shard degree: 1~4
>
> | Method | Update actor runtime (s) | Ref log prob runtime (s) | Old log prob runtime (s) | Data loading latency (s) | Step elapsed_time (s) | Overall Throughput (tokens/s) |
> | --- | --- | --- | --- | --- | --- | --- |
> | veRL + Bucket Balancing (Baseline) | 13.2 (1.00x) | 9.07 (1.00x) | 9.09 (1.00x) | 184.6 (1.00x) | 274.6 (1.00x) | 1977 (1.00x) |
> | **FlexRL w/ speedup** | **5.39 (2.48x)** | **3.79 (2.39x)** | **3.63 (2.50x)** | **6.598 (27.9x)** | **34.03 (8.05x)** | **15936(8.06x)** |
>
> Under this balanced workload, FlexRL still yields large gains over veRL: data-loading latency drops from 184.6 s to 6.6 s (27.9×), step time improved by 8.05×, and end-to-end throughput improves by 8.06×. Because computation aligns, the improvement mainly comes from eliminating data transfer overhead.
>
> In summary, evaluation on balanced workloads demonstrates that FlexRL is beneficial not only under highly skewed batches: the decentralized data pipeline removes the centralized multimodal I/O bottleneck.

---

> ### Author Response · Authors · 2025-11-25
> **Rebuttal by Authors Part 4**
>
> >### Q4: Performance under different settings.
>
> Thank you for this important question. To provide a comprehensive evaluation, we conducted experiments across a range of configurations: **batch size**, **sequence length** (`max_frame`), and **SP sharding range**. These parameters are critical in multimodal RL scenarios, as they directly impact memory usage, communication overhead, and load balancing efficiency. Below, we present our experimental setup and results.
>
> We evaluate FlexRL’s performance using different combinations of the above parameters. The evaluation setting is as follows
>
> - **Model**: MiMo-VL-7B (64 attention heads rather than 40 of QwenVL).
> - **Hardware**: 32 H800 GPUs
> - **Dataset:** Geo3K + NExTQA + LongVILA-Reason.
>     - Sampling weight: [5:2:1]
>     - `max_response_length` : 256.
> - Feature and parallelism:
>     - **Rollout model**: TP size=4
>     - FSDP size = 32
>     - FlexRL w/ both decentralized data pipeline and hybrid sequence sharding enabled.
>     - `max_sp` means the maximum sharding degree of **HSS**
>
> | Setting (max_sp, global_batch, max_frame) | Update actor runtime (s) | Ref log prob runtime (s) | Old log prob runtime (s) | Overall Throughput (tokens/s) |
> | --- | --- | --- | --- | --- |
> | `max_sp=2, global_batch=128, max_frame=64` | 3.9 | 1.4 | 2.5 | 5856 |
> | `max_sp=4, global_batch=256, max_frame=128` | 8.1 | 2.6 | 4.4 | 7266 |
> | `max_sp=4, global_batch=256, max_frame=512` | 8.9 | 2.8 | 4.1 | 10154 |
> | `max_sp=4, global_batch=384, max_frame=256` | 11.4 | 4.1 | 7.8 | 8255 |
> | `max_sp=8, global_batch=384, max_frame=256` | 10.9 | 3.7 | 7.6 | 8326 |
> | `max_sp=8, global_batch=192, max_frame=512` | 9.5 | 3.2 | 6.0 | 9050 |
>
> We observe that the system scales well with batch size and sequence length, demonstrating FlexRL's efficiency in computation-bound regimes. Increasing the maximum sharding degree (`max_sp`) delivers higher performance due to higher scheduling space and does not incur significant communication overhead. In summary, FlexRL maintains high throughput across a wide range of `max_frame` settings, effectively handling the load imbalance challenges of VLM RL training.
>
> >### Q5: Code release or community usage feedback.
>
> Thank you for your suggestion of releasing the code, which is important for an MLSys contribution. We have provided an anonymous version of the code [Anonymous Repository](https://anonymous.4open.science/r/rebuttal-8182), which currently includes the core implementation and startup scripts for the main experiments. FlexRL has been deployed in our production VLM training and significantly improve training efficiency and GPU utilization in our cluster. We are also actively refactoring the code and will release a complete opensource version after passing the internal review.
>
> We hope our responses have addressed your concerns, and we look forward to any further questions you may have!

---

### Author Response · Authors · 2025-12-02
**Rebuttal Summary**

We sincerely thank all reviewers and the AC for their thoughtful and constructive feedback! We provide a brief summary of our paper and rebuttal:

1. **Rating Updates**
    - Reviewer cr9u (**raise score**): Explicitly confirmed  that "*concerns have been addressed*" and stated "*I will raise my score*"  (posted on Nov 25)
    - Reviewer apGj & uQAx (no response): Their main concerns (lack of baselines/ablations) have been fully addressed with new experiments. We believe our response warrants a positive re-evaluation.
2. **Recognized Strengths.**
Reviewers have consistently acknowledged the system's value and novelty:
    - **Novelty:** Reviewers highlighted the *Hybrid Sequence Sharding* mechanism as "ingenious" and "novel", offering a promising direction for handling workload skew.
    - **Systematic Analysis:** The paper was praised by all reviewers for accurately identifying key systemic bottlenecks and demonstrating "strong system-level completeness".
    - **Performance:** Significantly outperforms existing frameworks on vision RL training tasks, achieving up to **7.66x** end-to-end throughput improvement.
3. **Key Responses.**
In response to reviewer requests, we added substantial experimental evidence. Main experiments are as follows:
    - **Outperforming SOTA**: Compared against LongRL (NeurIPS 2025), FlexRL achieves 5.32x speedup due to superior handling of imbalanced workloads.
    - **Comprehensive E2E evaluation & Ablation Studies**: We provided a detailed breakdown showing the individual contributions of the Decentralized Data Pipeline (removing I/O bottlenecks) and Hybrid Sequence Sharding (balancing compute) across all stages and E2E throughput.
    - **Rigorous System Analysis:** Provided the formal cost estimation model, detailed scheduling heuristic, and quantitative communication overhead analysis.
    - **Convergence Verification**: We provided training curves demonstrating that FlexRL improves efficiency without compromising model convergence/accuracy.

---

### Meta-Review · Area_Chair_PdRi · 2026-01-07

**Summary:**

The reviewers raised some concerns related to the manuscript, which include missing ablations (apGj,uQAx), missing trade-offs and individual contributions of elements constituting the whole pipeline (apGj, uQAx), missing runtime and accuracy (cr9u), and key details missing (uQAx). The evaluation from the reviewers is of two marginally above the acceptance and one marginally below. Notably, the confidence expressed by two of the three reviewers is low, while one marginally above expresses higher confidence.

**Reviewer Concerns:**

In their long rebuttal, the authors provided multiple extra elements to their work. Notably, the inclusion of the ablation study and the comparison against another method such as VeRL/LongRL were essential missing elements that must be included in the revised manuscript. Still, not providing average reward and analysis of the performing model on existing benchmarks limits the assessment behind the quality of the achieved solution - the linked anonymous figure shows that the average reward is in the same order as competing methods (but being more efficient); still, a more systematic report of this is still missing.

**Reviewer Scores:**

Overall, it is realistic to assume that, after an exchange where the reviewers would ask for consistently report the mean critic score for all the tested approaches to show that the score remains in the same magnitude (and one possibility is to average the last 50 steps reporting std), the evaluation would be aligned to a marginal acceptance. The authors are required to update the paper with this and with the extra elements provided at rebuttal if the paper is accepted.

---

### Decision · Program_Chairs · 2026-01-26

Accept (Poster)